# Continuous-time Models
# for Stochastic Optimization Algorithms

**Antonio Orvieto**
Department of Computer Science
ETH Zurich, Switzerland *

**Aurelien Lucchi**
Department of Computer Science
ETH Zurich, Switzerland

## Abstract

We propose new continuous-time formulations for first-order stochastic optimization algorithms such as mini-batch gradient descent and variance-reduced methods. We exploit these continuous-time models, together with simple Lyapunov analysis as well as tools from stochastic calculus, in order to derive convergence bounds for various types of non-convex functions. Guided by such analysis, we show that the same Lyapunov arguments hold in discrete-time, leading to matching rates. In addition, we use these models and Itô calculus to infer novel insights on the dynamics of SGD, proving that a decreasing learning rate acts as time warping or, equivalently, as landscape stretching.

## 1 Introduction

We consider the problem of finding the minimizer of a smooth non-convex function $f : \mathbb{R}^d \to \mathbb{R}$: $x^* := \arg\min_{x \in \mathbb{R}^d} f(x)$. We are here specifically interested in a finite-sum setting which is commonly encountered in machine learning and where $f(\cdot)$ can be written as a sum of individual functions over datapoints. In such settings, the optimization method of choice is mini-batch Stochastic Gradient Descent (MB-SGD) which simply iteratively computes stochastic gradients based on averaging from sampled datapoints. The advantage of this approach is its cheap per-iteration complexity which is independent of the size of the dataset. This is of course especially relevant given the rapid growth in the size of the datasets commonly used in machine learning applications. However, the steps of MB-SGD have a high variance, which can significantly slow down the speed of convergence [22, 36]. In the case where $f(\cdot)$ is a strongly-convex function, SGD with a decreasing learning rate achieves a sublinear rate of convergence in the number of iterations, while its deterministic counterpart (i.e. full Gradient Descent, GD) exhibits a linear rate of convergence.

There are various ways to improve this rate. The first obvious alternative is to systematically increase the size of the mini-batch at each iteration: [20] showed that a controlled increase of the mini-batch size yields faster rates of convergence. An alternative, that has become popular recently, is to use variance reduction (VR) techniques such as SAG [56], SVRG [32], SAGA [16], etc. The high-level idea behind such algorithms is to re-use past gradients on top of MB-SGD in order to reduce the variance of the stochastic gradients. This idea leads to faster rates: for general $L$-smooth objectives, both SVRG and SAGA find an $\epsilon$-approximate stationary point[2] in $\mathcal{O}\left(Ln^{2/3}/\epsilon\right)$ stochastic gradient computations [3, 53], compared to the $\mathcal{O}\left(Ln/\epsilon\right)$ needed for GD [45] and the $\mathcal{O}\left(1/\epsilon^2\right)$ needed for MB-SGD [22]. As a consequence, most modern state-of-the-art optimizers designed for general smooth objectives (Natasha [2], SCSG [37], Katyusha [1], etc) are based on such methods. The optimization algorithms discussed above are typically analyzed in their discrete form. One alternative that has recently become popular in machine learning is to view these methods as continuous-time

processes. By doing so, one can take advantage of numerous tools from the field of differential equations and stochastic calculus. This has led to new insights about non-trivial phenomena in non-convex optimization [40, 31, 60] and has allowed for more compact proofs of convergence for gradient methods [57, 42, 34]. This perspective appears to be very fruitful, since it also has led to the development of new discrete algorithms [68, 9, 64, 65]. Finally, this connection goes beyond the study of algorithms, and can be used for neural network architecture design [14, 12].

This success is not surprising, given the impact of continuous-time models in various scientific fields including, e.g., mathematical finance, where these models are often used to get closed-form solutions for derivative prices that are not available for discrete models (see e.g. the celebrated Black-Scholes formula [10], which is derived from Itô's lemma [30]). Many other success stories come from statistical physics [18], biology [24] and engineering. Nonetheless, an important question, which has encouraged numerous debates (see e.g. [62]), is about the reason behind the effectiveness of continuous-time models. In optimization, this question is partially addressed for deterministic accelerated methods by the works of [63, 9, 57] that provide a link between continuous and discrete time. However, we found that this problem has received less attention in the context of stochastic non-convex optimization and does not cover recent developments such as [32]. We therefore focus on the latter setting for which we provide detailed comparisons and analysis of continuous- and discrete-time methods. The paper is organized as follows:

1. In Sec. 2 we build new continuous-time models for SVRG and mini-batch SGD — which include the effect of decaying learning rates and increasing batch-sizes. We show existence and uniqueness of the solution to the corresponding stochastic differential equations.
2. In Sec. 3.1 we derive novel and interpretable non-asymptotic convergence rates for our models, using the elegant machinery provided by stochastic calculus. We focus on various classes of non-convex functions relevant for machine learning (see list in Sec. 3).
3. In Sec. 3.2 we complement each of our rates in continuous-time with equivalent results for the algorithmic counterparts, using the same Lyapunov functions. This shows an *algebraic equivalence* between continuous and discrete time and proves the effectiveness of our modeling technique. To the best of our knowledge, most of these rates (in full generality) are novel [3].
4. In Sec. 4.1 we provide a new interpretation for the distribution induced by SGD with decreasing stepsizes based on the Øksendal's time change formula — which reveals an underlying time warping phenomenon that can be used for designing Lyapunov functions.
5. In Sec. 4.2 we provide a dual interpretation of this last phenomenon as landscape stretching.

At a deeper level, our work proves that continuous-time models can adequately guide the analysis of stochastic gradient methods and provide new thought-provoking perspectives on their dynamics.

## 2  Unified models of stochastic gradient methods

Let $\{f_i\}_{i=1}^N$ be a collection of functions s.t. $f_i : \mathbb{R}^d \to \mathbb{R}$ for any $i \in [N]$ and $f(\cdot) := \frac{1}{N} \sum_{i=1}^N f_i(\cdot)$. In order to minimize $f(\cdot)$, first-order stochastic optimization algorithms rely on some noisy (but usually unbiased) estimator $\mathcal{G}(\cdot)$ of the gradient $\nabla f(\cdot)$. In its full generality, Stochastic Gradient Descent (SGD) builds a sequence of estimates of the solution $x^*$ in a recursive way:

$$x_{k+1} = x_k - \eta_k \mathcal{G}\left(\{x_i\}_{0 \le i \le k}, k\right), \tag{SGD}$$

where $(\eta_k)_{k \ge 0}$ is a non-increasing deterministic sequence of positive numbers called the *learning rates sequence*. Since $\mathcal{G}(x_k, k)$ is stochastic, $\{x_k\}_{k \ge 0}$ is a stochastic process on some countable probability space $(\Omega, \mathcal{F}, \mathbb{P})$. Throughout this paper, we denote by $\{\mathcal{F}_k\}_{k \ge 0}$ the natural filtration induced by $\{x_k\}_{k \ge 0}$; by $\mathbb{E}$ the expectation over all the information $\mathcal{F}_\infty$ and by $\mathbb{E}_{\mathcal{F}_k}$ the conditional expectation given the information at step $k$. We consider the two following popular designs for $\mathcal{G}(\cdot)$.

**i) MB gradient estimator.** The mini-batch gradient estimator at iteration $k$ is $\mathcal{G}_{\mathrm{MB}}(x_k, k) := \frac{1}{b_k} \sum_{i_k \in \Omega_k} \nabla f_{i_k}(x_k)$, where $b_k := |\Omega_k|$ and the elements of $\Omega_k$ (the *mini-batch*) are sampled at each iteration $k$ independently, uniformly and with replacement from $[N]$. Since $\Omega_k$ is random, $\mathcal{G}_{\mathrm{MB}}(x)$ is a random variable with conditional (i.e. taking out randomness in $x_k$) mean and covariance

$$\mathbb{E}_{\mathcal{F}_{k-1}}\left[\mathcal{G}_{\mathrm{MB}}(x_k, k)\right] = \nabla f(x_k), \qquad \mathbb{Cov}_{\mathcal{F}_{k-1}}\left[\mathcal{G}_{\mathrm{MB}}(x_k, k)\right] = \frac{\Sigma_{\mathrm{MB}}(x_k)}{b_k}, \tag{1}$$

where $\Sigma_{\text{MB}}(x) := \frac{1}{N} \sum_{i=1}^{N} (\nabla f(x) - \nabla f_i(x)) (\nabla f(x) - \nabla f_i(x))^T$ is the one-sample covariance.

**ii) VR gradient estimator.** The basic idea of the original SVRG algorithm introduced in [32] is to compute the full gradient at some chosen *pivot* point and combine it with stochastic gradients computed at subsequent iterations. Combined with mini-batching [53], this gradient estimator is:

$$\mathcal{G}_{\text{VR}}(x_k, \tilde{x}_k, k) := \frac{1}{b_k} \sum_{i_k \in \Omega_k} \nabla f_{i_k}(x_k) - \nabla f_{i_k}(\tilde{x}_k) + \nabla f(\tilde{x}_k),$$

where $\tilde{x}_k \in \{x_0, x_1, \dots, x_{k-1}\}$ is the pivot used at iteration $k$. This estimator is unbiased, i.e. $\mathbb{E}_{\mathcal{F}_{k-1}}[\mathcal{G}_{\text{VR}}(x_k, \tilde{x}_k, k)] = \nabla f(x_k)$. Its covariance is $\mathbb{C}\text{ov}_{\mathcal{F}_{k-1}}[\mathcal{G}_{\text{VR}}(x_k, \tilde{x}_k, k)] = \frac{\Sigma_{\text{VR}}(x_k, \tilde{x}_k)}{b_k}$ with $\Sigma_{\text{VR}}(x, y) := \frac{1}{N} \sum_{i=1}^{N} (\nabla f_i(x) - \nabla f_i(y) + \nabla f(y) - \nabla f(x)) (\nabla f_i(x) - \nabla f_i(y) + \nabla f(y) - \nabla f(x))^T$.

### 2.1 Building the perturbed gradient flow model

We take inspiration from [38] and [27] and build continuous-time models for SGD with either the MB or the SVRG gradient estimators. The procedure has three steps.

(S1) We first define the *discretization stepsize* $h := \eta_0$ — this variable is essential to provide a link between continuous and discrete time. We assume it to be fixed for the rest of this subsection. Next, we define the *adjustment-factors sequence* $(\psi_k)_{k \geq 0}$ s.t. $\psi_k = \eta_k/h$ (cf. Eq. 9 in [38]). In this way — we decouple the two information contained in $\eta_k$: $h$ controls the overall size of the learning rate and $\psi_k$ handles its variation[4] during training.

(S2) Second, we write SGD as $x_{k+1} = x_k - \eta_k(\nabla f(x_k) + V_k)$, where the error $V_k$ has mean zero and covariance $\Sigma_k$. Next, let $\Sigma_k^{1/2}$ be the principal square root[5] of $\Sigma_k$, we can write SGD as

$$x_{k+1} = x_k - \eta_k \nabla f(x_k) - \eta_k \Sigma_k^{1/2} Z_k, \tag{PGD}$$

where $Z_k$ is a random variable with zero mean and unit covariance[6]. In order to build *simple* continuous-time models, we assume that each $Z_k$ is Gaussian distributed: $Z_k \sim \mathcal{N}(0_d, I_d)$. To highlight this assumption, we will refer to the last recursion as Perturbed Gradient Descent (PGD) [15]. In Sec. 2.1 we motivate why this assumption, which is commonly used in the literature [38], is not restrictive for our purposes. By plugging in either $\Sigma_k = \Sigma_{\text{MB}}(x_k)/b_k$ or $\Sigma_k = \Sigma_{\text{VR}}(x_k, \tilde{x}_k)/b_k$, we get a discrete model for SGD with the MB or VR gradient estimators.

(S3) Finally, we lift these PGD models to continuous time. The first step is to rewrite them using $\psi_k$:

$$x_{k+1} = x_k - \underbrace{\psi_k \nabla f(x_k)}_{\text{adjusted gradient drift}} h + \underbrace{\psi_k \sqrt{h/b_k} \, \sigma_{\text{MB}}(x_k)}_{\text{adjusted mini-batch volatility}} \sqrt{h} Z_k \tag{MB-PGD}$$

$$x_{k+1} = x_k - \underbrace{\psi_k \nabla f(x_k)}_{\text{adjusted gradient drift}} h + \underbrace{\psi_k \sqrt{h/b_k} \, \sigma_{\text{VR}}(x_k, x_{k-\xi_k})}_{\text{adjusted variance-reduced volatility}} \sqrt{h} Z_k \tag{VR-PGD}$$

where $\sigma_{\text{MB}}(x) := \Sigma^{1/2}(x)$, $\sigma_{\text{VR}}(x, y) := \Sigma_{\text{VR}}^{1/2}(x, y)$ and $\xi_k \in [k]$ quantifies the pivot staleness. Readers familiar with stochastic analysis might recognize that MB-PGD and VR-PGD are the steps of a numerical integrator (with stepsize $h$) of an SDE and of an SDDE, respectively. For convenience of the reader, we give an hands-on introduction to these objects in App. B.

The resulting continuous-time models, which we analyse in this paper, are

$$dX(t) = -\psi(t)\nabla f(X(t)) \, dt + \psi(t)\sqrt{h/b(t)} \, \sigma_{\text{MB}}(X(t)) \, dB(t) \tag{MB-PGF}$$

$$dX(t) = -\psi(t)\nabla f(X(t)) \, dt + \psi(t)\sqrt{h/b(t)} \, \sigma_{\text{VR}}(X(t), X(t - \xi(t))) \, dB(t) \tag{VR-PGF}$$

where

- $\xi : \mathbb{R}_+ \to [0, \mathfrak{T}]$, the *staleness function*, is s.t. $\xi(hk) = \xi_k$ for all $k \geq 0$;
- $\psi(\cdot) \in \mathcal{C}^1(\mathbb{R}_+, [0,1])$, the *adjustment function*, is s.t. $\psi(hk) = \psi_k$ for all $k \geq 0$ and $\frac{d\psi(t)}{dt} \leq 0$;
- $b(\cdot) \in \mathcal{C}^1(\mathbb{R}_+, \mathbb{R}_+)$, the *mini-batch size function* is s.t. $b(hk) = b_k$ for all $k \geq 0$ and $b(t) \geq 1$;
- $\{B(t)\}_{t \geq 0}$ is a $d-$dimensional Brownian Motion on some filtered probability space.

We conclude this subsection with some important remarks and clarifications on the procedure above.

**On the Gaussian assumption.** In (S2) we assumed that $Z_k$ is Gaussian distributed. If the mini-batch size $b_k$ is large enough and the gradients are sampled from a distribution with finite variance, then the assumption is sound: indeed, by the Berry–Esseen Theorem (see e.g. [17]), $Z_k$ approaches $\mathcal{N}(0_d, I_d)$ in distribution with a rate $\mathcal{O}\left(1/\sqrt{b_k}\right)$. However, if $b_k$ is small or the underlying variance is unbounded, the distribution of $Z_k$ has heavy tails [58]. Nonetheless, in the large-scale optimization literature, the gradient variance is generally assumed to be bounded (see e.g. [22], [11]) — hence, we keep this assumption, which is practical and reasonable for many problems (likewise assumed in the related literature [51, 42, 34, 38, 39]). Also, taking a different yet enlightening perspective, it is easy to see that (see Sec. 4 of [11]), if one cares only about expected convergence guarantees — only the first and the second moments of the stochastic gradients have a quantitative effect on the rate.

**Approximation guarantees.** Recently, [28, 39] showed that for a special case of MB-PGF ($\psi_k = 1$, and $b_k$ constant), its solution $\{X(t)\}_{0 \leq t \leq T}$ compares to SGD as follows: let $K = \lfloor T/h \rfloor$ and consider the iterates $\{x_k\}_{k \in [K]}$ of mini-batch SGD (i.e. *without* Gaussian assumption) with fixed learning rate $h$. Under mild assumptions on $f(\cdot)$, there exists a constant $C$ (independent of $h$) such that $\|\mathbb{E}[x_k] - \mathbb{E}[X(kh)]\| \leq Ch$ for all $k \in [K]$. Their proof argument relies on semi-group expansions of the solution to the Kolmogorov backward equation, and can be adapted to provide a similar result for our (more general) equations. However, this approach to motivate the continuous-time formulation is *very limited* — as $C$ depends exponentially on $T$ (see also [57]). Nonetheless, under strong-convexity, some uniform-in-time (a.k.a. *shadowing*) results were recently derived in [48, 19]. In this paper, we take a different approach (similarly to [57] for deterministic methods) and provide instead matching convergence rates in continuous and in discrete time using the same Lyapunov function. We note that this is still a very strong indication of the effectiveness of our model to study SGD, since it shows an *algebraic equivalence* between the continuous and the discrete case.

**Comparison to the "ODE method".** A powerful technique in stochastic approximation [36] is to study SGD through the *deterministic* ODE $\dot{X} = -\nabla f(X)$. A key result is that SGD, with decreasing learning rate under the Robbins Monro [55] conditions, behaves like this ODE in the limit. Hence the ODE can be used to characterize the *asymptotic* behaviour of SGD. In this work we instead take inspiration from more recent literature [39] and build *stochastic* models which include the effect of a decreasing learning rate into the drift and the volatility coefficients through the adjustment function $\psi(\cdot)$. This allows, in contrast to the ODE method[7], to provide *non-asymptotic* arguments and convergence rates.

**Local minima width.** Our models confirm, as noted in [31], that the ratio of (initial) learning rate $h$ to batch size $b(t)$ is a determinant factor of SGD dynamics. Compared to [31], our model is more general: indeed, we will see in Sec. 4.2 that the adjustment function also plays a fundamental role in determining the width of the final minima — since it acts like a "function stretcher".

## 2.2 Existence and uniqueness

Prior works that take an approach similar to ours [35, 27, 42], assume the one-sample volatility $\sigma(\cdot)$ to be Lipschitz continuous. This makes the proof of existence and uniqueness straightforward (cf. a textbook like [41]), but we claim such assumption is not trivial in our setting where $\sigma(\cdot)$ is data-dependent. Indeed, $\sigma(\cdot)$ is the result of a square root operation on the gradient covariance — and the square root function is not Lipschitz around zero. App. C is dedicated to a rigorous proof of existence and uniqueness, which is verified under the following condition:

**(H)**     Each $f_i(\cdot)$ is $\mathcal{C}^3$, with bounded third derivative and $L$-smooth.

This hypothesis is arguably not restrictive as it is usually satisfied by many loss functions encountered in machine learning. As a result, under **(H)**, with probability 1 the realizations of the stochastic processes $\{f(X(t))\}_{t>0}$ and $\{X(t)\}_{t>0}$ are continuous functions of time.

## 3   Matching convergence rates in continuous and discrete time

Even though in optimization, convex functions are central objects of study, many interesting objectives found in machine learning are non-convex. However, most of the time, such functions still exhibit some regularity. For instance, [25] showed that linear LSTMs induce weakly-quasi-convex objectives.

**(H$_{\text{WQC}}$)**   $f(\cdot)$ is $\mathcal{C}^1$ and exists $\tau > 0$ and $x^\star$ s.t. $\langle \nabla f(x), x - x^\star \rangle \geq \tau(f(x) - f(x^\star))$ for all $x \in \mathbb{R}^d$.

Intuitively, **(H$_{\text{WQC}}$)** requires the negative gradient to be always aligned with the direction of a global minimum $x^\star$. Convex differentiable functions are weakly-quasi-convex (with $\tau = 1$), but the WQC class is richer and actually allows functions to be locally concave. Another important class of problems (e.g., under some assumptions, matrix completion [61]) satisfy the Polyak-Łojasiewicz property, which is the weakest known sufficient condition for GD to achieve linear convergence [50].

**(H$_{\text{PL}}$)**   $f(\cdot)$ is $\mathcal{C}^1$ and there exists $\mu > 0$ s.t. $\|\nabla f(x)\|^2 \geq 2\mu(f(x) - f(x^\star))$ for all $x \in \mathbb{R}^d$.

One can verify that if $f(\cdot)$ is strongly-convex, then it is PŁ. However, PŁ functions are not necessarily convex. What's more, a broad class of problems (dictionary learning [5], phase retrieval [13], two-layer MLPs [39]) are related to a stronger condition: the restricted-secant-inequality [66].

**(H$_{\text{RSI}}$)**   $f(\cdot)$ is $\mathcal{C}^1$ and there exists $\mu > 0$ s.t. $\langle \nabla f(x), x - x^* \rangle \geq \frac{\mu}{2}\|x - x^*\|^2$ for all $x \in \mathbb{R}^d$.

In [33] the authors prove strong-convexity $\Rightarrow$ **(H$_{\text{RSI}}$)** $\Rightarrow$ **(H$_{\text{PL}}$)** (with different constants).

### 3.1   Continuous-time analysis

First, we derive non-asymptotic rates for MB-PGF. For convenience, we define $\varphi(t) := \int_0^t \psi(s)ds$, which plays a fundamental role (see Sec. 4.1). As [42, 34], we introduce a bound on the volatility.

**(H$\sigma$)** $\sigma_*^2 := \sup_{x \in \mathbb{R}^d} \|\sigma_{\text{MB}}(x)\sigma_{\text{MB}}(x)^T\|_s < \infty$, where $\|\cdot\|_s$ denotes the spectral norm.

---

**Theorem 1.** *Assume (H), (H$\sigma$). Let $t > 0$ and $\tilde{t} \in [0, t]$ be a random time point with distribution $\frac{\psi(\tilde{t})}{\varphi(t)}$ for $\tilde{t} \in [0, t]$ (and $0$ otherwise). The solution to __MB-PGF__ is s.t.*

$$\mathbb{E}\left[\|\nabla f(X(\tilde{t}))\|^2\right] \leq \frac{f(x_0) - f(x^\star)}{\varphi(t)} + \frac{h\,d\,L\,\sigma_*^2}{2\,\varphi(t)}\int_0^t \frac{\psi(s)^2}{b(s)}ds.$$

---

*Proof.* We use the energy function $\mathcal{E}(x, t) := f(x) - f(x^\star)$. Details in App. D.2.   ∎

---

**Theorem 2.** *Assume (H), (H$\sigma$), (H$_{\text{WQC}}$). Let $\tilde{t}$ be as in Thm. 1. The solution to __MB-PGF__ is s.t.*

$$\mathbb{E}\left[f(X(\tilde{t})) - f(x^\star)\right] \leq \frac{\|x_0 - x^\star\|^2}{2\,\tau\,\varphi(t)} + \frac{h\,d\,\sigma_*^2}{2\,\tau\,\varphi(t)}\int_0^t \frac{\psi(s)^2}{b(s)}ds \tag{W1}$$

$$\mathbb{E}\left[(f(X(t)) - f(x^\star))\right] \leq \frac{\|x_0 - x^\star\|^2}{2\,\tau\,\varphi(t)} + \frac{h\,d\,\sigma_*^2}{2\,\tau\,\varphi(t)}\int_0^t (L\,\tau\,\varphi(s) + 1)\frac{\psi(s)^2}{b(s)}ds. \tag{W2}$$

---

*Proof.* We use the energy functions $\mathcal{E}_1, \mathcal{E}_2$ s.t. $\mathcal{E}_1(x) := \frac{1}{2}\|x - x^\star\|^2$ and $\mathcal{E}_2(x, t) := \tau\varphi(t)(f(x)) - f(x^\star)) + \frac{1}{2}\|x - x^\star\|^2$ for (W1) and (W2), respectively. Details in App. D.2.   ∎

---

**Theorem 3.** *Assume (H), (H$\sigma$), (H$_{\text{PL}}$). The solution to __MB-PGF__ is s.t.*

$$\mathbb{E}[f(X(t)) - f(x^\star)] \leq e^{-2\mu\varphi(t)}(f(x_0) - f(x^\star)) + \frac{h\,d\,L\,\sigma_*^2}{2}\int_0^t \frac{\psi(s)^2}{b(s)}e^{-2\mu(\varphi(t)-\varphi(s))}ds.$$

---

Table 1: Asymptotic rates for MB-PGF under $\psi(t) = \mathcal{O}(t^{-a})$ in the form $\mathcal{O}(t^{-\beta})$. $\beta$ shown in the table as a function of $a$. "$\sim$" indicates randomization of the result. Rates *match* with Tb. 1 in [43].

| $a$ | (H), (H$\sigma$), (H$_{\text{PL}}$) Cor. 3 | (H), (H$\sigma$), (H$_{\text{WQC}}$) Cor. 2 | ($\sim$), (H), (H$\sigma$), (H$_{\text{WQC}}$) Cor. 2 | ($\sim$), (H), (H$\sigma$) Cor. 1 |
|---|---|---|---|---|
| (0 , 1/2) | $a$ | $\times$ | $a$ | $a$ |
| (1/2 , 2/3) | $a$ | $2a - 1$ | $1 - a$ | $1 - a$ |
| (2/3 , 1) | $a$ | $1 - a$ | $1 - a$ | $1 - a$ |

*Proof.* We use the energy function $\mathcal{E}(x, t) := e^{2\mu\varphi(t)}(f(x) - f(x^\star))$. Details in App. D.2. ∎

**Decreasing mini-batch size.** From Thm. 1, 2, 3, it is clear that, as it is well known [11, 6], a simple way to converge to a local minimizer is to pick $b(\cdot)$ increasing as a function of time. However, this corresponds to dramatically increasing the complexity in terms of gradient computations. In continuous-time, we can account for this by introducing $\beta(t) = \int_0^t b(s)ds$, proportional to the number of computed gradients at time $t$. The complexity in number of gradient computations can be derived by substituting into the final rate the *new time variable* $\beta^{-1}(t)$ instead of $t$. As we will see in Thm. 5, this concept extends to a more general setting and leads to valuable insights.

**Asymptotic rates.** Another way to guarantee convergence to a local minimizer is to decrease $\psi(\cdot)$. In App. D.3 we derive asymptotic rates for $\psi(t) = \mathcal{O}(t^{-a})$ and report the results in Tb. 1. The results *match exactly* the corresponding know rates for SGD, stated under stronger assumptions in [43]. As for increasing $b(\cdot)$, decreasing $\psi(\cdot)$ can also be seen as performing a time warp (see Thm. 5).

**Ball of convergence.** For $\psi(t) = 1$, the sub-optimality gap derived in App. D.3.1 *matches* [11].

In contrast to $\mathcal{G}_{\text{MB}}(\cdot)$, [3, 4] have shown that significant speed-ups are hard to obtain from parallel gradient computations (i.e. for $b(t) > 1$) using $\mathcal{G}_{\text{VR}}(\cdot)$ [8]. Also, our results for MB-PGF as well as prior work [67, 3, 4, 53] suggest that linear rates can only be obtained with $\psi(t) = 1$. Hence, for our analysis of VR-PGF, we focus on the case $b(t) = \psi(t) = 1$. The following result, in the spirit of [32, 4], relates to the so-called *Option II* of SVRG.

---

**Theorem 4.** *Assume (H), (H$_{\text{RSI}}$) and choose $\xi(t) = t - \sum_{j=1}^{\infty} \delta(t - j\mathfrak{T})$ (sawtooth wave), where $\delta(\cdot)$ is the Dirac delta. Let $\{X(t)\}_{t \geq 0}$ be the solution to __VR-PGF__ with additional jumps at times $(j\mathfrak{T})_{j \in \mathbb{N}}$: we pick $X(j\mathfrak{T} + \mathfrak{T})$ uniformly in $\{X(s)\}_{j\mathfrak{T} \leq s < (j+1)\mathfrak{T}}$. Then,*

$$\mathbb{E}[\|X(j\mathfrak{T}) - x^\star\|^2] = \left(\frac{2hL^2\mathfrak{T} + 1}{\mathfrak{T}(\mu - 2hL^2)}\right)^j \|x_0 - x^*\|^2.$$

---

**Previous Literature (SDEs for MB-PGF).** [42] studied dual averaging using a similar SDE model in the convex setting, under vanishing and persistent volatility. Part of their results are similar, yet less general and not directly comparable. [51] studied a specific case of our equations, under constant volatility (see also [52] and references therein). [34, 65, 64] studied extentions to [42] including acceleration [45] and AC-SA [21]. To the best of our knowledge, there hasn't been yet any analysis of SVRG in continuous-time in the literature.

### 3.2 Discrete-time analysis and algebraic equivalence

We provide matching algorithmic counterparts (using the same Lyapunov function) for all our non-asymptotic rates in App. D, along with Tb. 2 to summarize the results. We stress that the rates we prove in discrete-time (i.e. for SGD with gradient estimators $\mathcal{G}_{\text{MB}}$ or $\mathcal{G}_{\text{VR}}$) hold *without Gaussian noise assumption*. This is a key result of this paper, which indicates that the tools of Itô calculus [30] —which are able to provide more compact proofs [42, 52] — yield calculations which are *equivalent* to the ones used to analyze standard SGD. We invite the curious reader to go through the proofs in the appendix to appreciate this correspondence as well as to inspect Tb. 3 in the appendix, which provides a comparison of the discrere-time rates with Thms. 1, 2, 3 and 4.

| Cond. | Rate (Discrete-time, no Gaussian assumption) | Thm. |
|---|---|---|
| $(\sim)$,(H-),(H$\sigma$) | $\dfrac{2\,(f(x_0) - f(x^\star))}{(h\varphi_{k+1})} + \dfrac{h\,d\,L\,\sigma_*^2}{(h\varphi_{k+1})} \sum_{i=0}^{k} \dfrac{\psi_i^2}{b_i} h$ | E.1 |
| $(\sim)$,(H-),(H$\sigma$),(HWQC) | $\dfrac{\|x_0 - x^\star\|^2}{\tau\,(h\varphi_{k+1})} + \dfrac{d\,h\,\sigma_*^2}{\tau\,(h\varphi_{k+1})} \sum_{i=0}^{k} \dfrac{\psi_i^2}{b_i} h$ | E.2 |
| (H-),(H$\sigma$),(HWQC) | $\dfrac{\|x_0 - x^\star\|^2}{2\,\tau\,(h\varphi_{k+1})} + \dfrac{h\,d\,\sigma_*^2}{2\,\tau\,(h\varphi_{k+1})} \sum_{i=0}^{k} (1 + \tau\varphi_{i+1}L) \dfrac{\psi_i^2}{b_i} h$ | E.2 |
| (H-),(H$\sigma$),(HPŁ) | $\prod_{i=0}^{k}(1 - \mu\,h\psi_i)(f(x_0) - f(x^\star)) + \dfrac{h\,d\,L\,\sigma_*^2}{2} \sum_{i=0}^{k} \dfrac{\prod_{\ell=0}^{k}(1 - \mu\,h\psi_\ell)}{\prod_{j=0}^{i}(1 - \mu\,h\psi_l)} \dfrac{\psi_i^2}{b_i} h$ | E.3 |
| (H-),(HRSI) | $\left(\dfrac{1 + 2L^2h^2m}{hm(\mu - 2L^2h)}\right)^j \|x_0 - x^*\|^2$  (under variance reduction) | E.4 |

Table 2: Summary of the rates we show in the appendix for SGD with mini-batch and VR, using a Lyapunov argument inspired by the continuous-time analysis. $(\sim)$ indicates randomized output. The reader should compare the results with Thms. 1, 2, 3, 4 (explicit comparison in the first page of the appendix). For the definition of the quantities in the rates, check App. E.

Now we ask the simple question: *why is this the case?* Using the concept of derivation from abstract algebra, in App. A.2 we show that the discrete difference operator and the derivative operator enjoy similar algebraic properties. *Crucially*, this is due to the smoothness of the underlying objective — which implies a chain-rule[9] for the difference operator. Hence, this equivalence is tightly linked with optimization and might lead to numerous insights. We leave the exploration of this fascinating direction to future work.

**Literature comparison (algorithms).** Even though partial[10] results have been derived for the function classes described above in [25, 54], an in-depth non-asymptotic analysis was still missing. Rates in Tb. 3 (stated above in continuous-time as theorems) provide a generalization to the results of [43] to the weaker function classes we considered (we *never* assume convexity). Regarding SVRG, the rate we report uses a proof similar[11] to [4, 53] and is comparable to [32] (under convexity).

## 4 Insights provided by continuous-time models

Building on the tools we used so far, we provide novel insights on the dynamics of SGD. First, in order to consider both MB-PGF and VR-PGF at the same time, we introduce a stochastic[12] matrix process $\{\sigma(t)\}_{t\geq 0}$ adapted to the Brownian motion:

$$dX(t) = -\psi(t)\nabla f(X(t))\,dt + \psi(t)\sqrt{h/b(t)}\sigma(t)\,dB(t). \qquad \text{(PGF)}$$

We show that annealing the learning rate through a decreasing $\psi(\cdot)$ can be viewed as performing a *time dilation* or, alternatively, as directly *stretching the objective function*. This view is inspired from the use of Girsanov theorem [23] in finance: a deep result in stochastic analysis which is the formal concept underlying the change of measure from real world to "risk-neutral" world.

## 4.1 Time stretching through Øksendal's formula

We notice that, in Thm. 1,2,3, the time variable $t$ is *always* filtered through the map $\varphi(\cdot)$. Hence, $\varphi(\cdot)$ seems to act as a new time variable. We show this rigorously using Øksendal's time change formula.

> **Theorem 5.** *Let $\{X(t)\}_{t \geq 0}$ satisfy PGF and define $\tau(\cdot) = \varphi^{-1}(\cdot)$, where $\varphi(t) = \int_0^t \psi(s)ds$. For all $t \geq 0$, $X(\tau(t)) = Y(t)$ in distribution, where $\{Y(t)\}_{t \geq 0}$ has the stochastic differential*
>
> $$dY(t) = -\nabla f(Y(t))dt + \sqrt{h\,\psi(\tau(t))/b(\tau(t))}\sigma(\tau(t))\,dB(t).$$

*Proof.* We use the substitution formula for deterministic integrals combined with Øksendal's formula for time change in stochastic integrals — a key result in SDE theory. Details in App. F. ∎

**Example.** We consider $b(t) = 1$, $\sigma(s) = \sigma I_d$ and $\psi(t) = 1/(t+1)$ (popular annealing procedure [11]); we have $\varphi(t) = \log(t+1)$ and $\tau(t) = e^t - 1$. $dX(t) = -\frac{1}{t+1}\nabla f(X(t))dt - \frac{\sqrt{h}\sigma}{t+1}dB(t)$ is s.t. the *sped-up solution* $Y(t) = X(e^t - 1)$ satisfies

$$dY(t) = -\nabla f(X(t))dt + \sqrt{h}\sigma e^{-t}dB(t). \qquad (2)$$

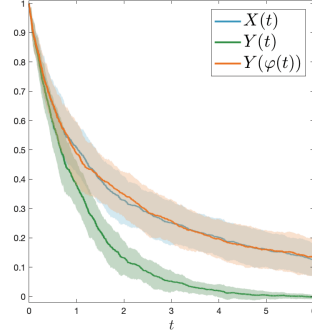

Figure 1: Verification of Thm. 5 on a 1d quadratic (100 samples): empirically $X(t) \stackrel{d}{=} Y(\varphi(t))$.

In the example, Eq. (2) is the model for SGD with constant learning rates but rapidly vanishing noise — which is arguably easier to study compared to the original equation, that also includes time-varying learning rates. Hence, this result draws a connection to SGLD [52] and to prior work on SDE models [42], which only considered $\psi(t) = 1$. But, most importantly — Thm. 5 allows for more *flexibility* in the analysis: to derive convergence rates[13] one could work with either $X$ (as we did in Sec. 2) or with $Y$ (and *slow-down* the rates afterwords).

We verify this result on a one dimensional quadratic, under the choice of parameters in our example, using Euler-Maruyama simulation (i.e. PGD) with $h = 10^{-3}$, $\sigma = 5$. In Fig. 1 we show the mean and standard deviation relative to 20 realization of the Gaussian noise.

Note that in the case of variance reduction, the volatility is decreasing as a function of time [3], even with $\psi(t) = 1$. Hence one gets a similar result without the change of variable.

## 4.2 Landspace stretching via solution feedback

Consider the (potentially non-convex) quadratic $f(x) = \langle x - x^\star, H(x - x^\star) \rangle$. WLOG we assume $x^\star = 0_d$ and that $H$ is diagonal. For simplicity, consider again the case $b(t) = 1$, $\sigma(s) = \sigma I_d$ and $\psi(t) = 1/(t+1)$. PGF reduces to a linear stochastic system:

$$dX(t) = -\frac{1}{t+1}HX(t)dt + \frac{h\sigma}{t+1}dB(t).$$

By the variation-of-constants formula [41], the expectation evolves without bias: $d\mathbb{E}[X(t)] = -\frac{1}{t+1}H\mathbb{E}[X(t)]dt$. If we denote by $u^i(t)$ the $i$-th coordinate of $\mathbb{E}[X(t)]$ we have $\frac{d}{dt}u^i(t) = -\frac{\lambda_i}{t+1}u^i(t)$, where $\lambda_i$ is the eigenvalue relative to the $i$-th direction. Using separation of variables, we find $u^i(t) = (t+1)^{-\lambda_i}u_0^i$. Moreover, we can invert space and time: $t = \left(u_0^i/u^i(t)\right)^{1/\lambda_i} - 1$. *Feeding back* this equation into the original differential — the system becomes autonomous:

$$\frac{d}{dt}u^i(t) = -\lambda_i(u_0^i)^{-\frac{1}{\lambda_i}}u^i(t)^{1+\frac{1}{\lambda_i}}.$$

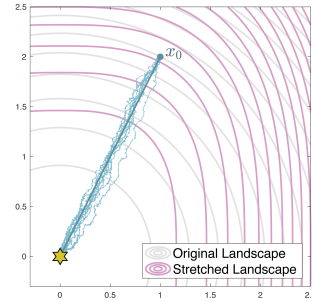

Figure 2: Landscape stretching for an isotropic paraboloid.

From this simple derivation we get two important insights on the dynamics of PGF:

1. Comparing the solution $u^i(t) = (t+1)^{-\lambda_i} u_0^i$ with the solution one would obtain with $\psi(t) = 1$, that is $e^{-\lambda_i t} u_0^i$ — we notice that the dynamics in the first case is much slower: we get polynomial convergence and divergence (when $\lambda_i \leq 0$) as opposed to exponential. This quantitatively shows that decreasing the learning rate could slow down (from exponential to polynomial) the dynamics of SGD around saddle points. However, note that, even though the speed is different, $u^i(\cdot)$ and $v^i(\cdot)$ move along the same path[14] by Thm 5.

2. Inspecting the equivalent formulation $\frac{d}{dt} u^i(t) = -\lambda_i (u_0^i)^{-\frac{1}{\lambda_i}} u^i(t)^{1+\frac{1}{\lambda_i}}$, we notice with surprise — that this is a gradient system. Indeed the RHS can be written as $C(\lambda_i, u_0^i) \nabla g_i(u^i(t))$, where $g_i(x) = x^{2+\frac{1}{\lambda_i}}$ is the *equivalent landscape* in the $i$-th direction. In particular, PGF on the simple quadratic $\frac{1}{2}\|x\|^2$ with learning rate decreasing as $1/t$ *behaves in expectation like PGF with constant learning rate on a cubic*. This shines new light on the fact that, as it is well known from the literature [44], by decreasing the learning rate we can only achieve sublinear convergence rates on strongly convex stochastic problems. From our perspective, this happens simply because the equivalent stretched landscape has vanishing curvature — hence, it is not strongly convex. We illustrate this last example in Fig. 2 and note that the stretching effect is tangent to the expected solution (in solid line).

We believe the landscape stretching phenomenon we just outlined to be quite general and to also hold asymptotically under strong convexity[15]: indeed it is well known that, by Taylor's theorem, in a neighborhood of the solution to a strongly convex problem the cost *behaves as its quadratic approximation*. In dynamical systems, this linearization argument can be made precise and goes under the name of Hartman-Grobman theorem (see e.g. [49]). Since the SDE we studied is memoryless (no momentum), at some point it will necessarily enter a neighborhood of the solution where the dynamics is described by result in this section. We leave the verification and formalization of the argument we just outlined to future research.

## 5 Conclusion

We provided a detailed comparisons and analysis of continuous- and discrete-time methods in the context of stochastic non-convex optimization. Notably, our analysis covers the variance-reduced method introduced in [32]. The continuous-time perspective allowed us to deliver new insights about how decreasing step-sizes lead to time and landscape stretching. There are many potential interesting directions for future research such as extending our analysis to mirror-descent or accelerated gradient-descent [35, 60], or to study state-of-the-art stochastic non-convex optimizers such as Natasha [2]. Finally, we believe it would be interesting to expand the work of [38, 39] to better characterize the convergence of MB-SGD and SVRG to the SDEs we studied here, perhaps with some asymptotic arguments similar to the ones used in mean-field theory [7, 8].

## Footnotes

*Correspondence to `orvietoa@ethz.ch`.

[2] A point $x$ where $\|\nabla f(x)\| \leq \epsilon$.

[3]We derive these rates in App. E and summarize them in Tb. 3.

[4] A popular choice (see e.g. [43]) is $\eta_k = Ck^{-\alpha}$, $\alpha \in [0, 1]$. Here, $h = C$ and $\psi_k = k^{-\alpha} \in [0, 1]$.

[5] The unique positive semidefinite matrix such that $\Sigma_k = \Sigma_k^{1/2} \Sigma_k^{1/2}$.

[6] Because $\Sigma_k^{1/2} Z_k$ has the same distribution as $V_k$, conditioned on $x_k$.

[7]This method is instead suitable to assess almost sure convergence and convergence in probability, which are not considered in this paper for the sake of delivering convergence rates for population quantities.

[8]See e.g. Thm. 7 in [53] for a counterexample.

[9]This is a key formula in the continuous-time analysis to compute the derivative of a Lyapunov function.

[10]The convergence under weak-quasi-convexity using a learning rate $C/\sqrt{k}$ and a randomized output is studied in [25] (Prop. 2.3 under Eq. 2.2 of their paper). On the same line, [33] studied the convergence for PŁusing a learning rate $C/\sqrt{k}$ and assuming bounded stochastic gradients. These results are strictly contained in our rates.

[11]In particular, the lack of convexity causes the factor $L^2$ in the linear rate.

[12]For MB-PGF, $\{\sigma(t)\}_{t\geq 0} := \{\sigma(X(t))\}_{t\geq 0}$. For VR-PGF, $\{\sigma(t)\}_{t\geq 0} := \{\sigma(X(t), X(t - \xi(t)))\}_{t\geq 0}$.

[13] The design of the Lyapunov function might be easier if we change time variable. This is the case in our setting, where $\varphi(t)$ comes directly into the Lyapunov functions and would be simply $t$ for the transformed SDE.

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
