[Supplementary Material · NeurIPS2019_SDEs_app.pdf]

# Appendix

## A  Summary of the rates derived in this paper

> *"A major task of mathematics today is to harmonize the continuous and the discrete, to include them in one comprehensive mathematics, and to eliminate obscurity from both."*
>
> – E.T. Bell, *Men of Mathematics*, 1937

| Cond. | Rate (Continuous-time) | Thm. |
|-------|------------------------|------|
| $(\sim)$,**(H-)**,**(H$\sigma$)** | $\dfrac{f(x_0)-f(x^\star)}{\varphi(t)} + \dfrac{h\,d\,L\,\sigma_*^2}{2\,\varphi(t)}\displaystyle\int_0^t \dfrac{\psi(s)^2}{b(s)}ds$ | 1 |
| $(\sim)$,**(H-)**,**(H$\sigma$)**,**(HWQC)** | $\dfrac{\|x_0-x^\star\|^2}{2\,\tau\,\varphi(t)} + \dfrac{h\,d\,\sigma_*^2}{2\,\tau\,\varphi(t)}\displaystyle\int_0^t \dfrac{\psi(s)^2}{b(s)}ds$ | 2 |
| **(H-)**,**(H$\sigma$)**,**(HWQC)** | $\dfrac{\|x_0-x^\star\|^2}{2\,\tau\,\varphi(t)} + \dfrac{h\,d\,\sigma_*^2}{2\,\tau\,\varphi(t)}\displaystyle\int_0^t (L\,\tau\,\varphi(s)+1)\dfrac{\psi(s)^2}{b(s)}ds$ | 2 |
| **(H-)**,**(H$\sigma$)**,**(HPŁ)** | $e^{-2\mu\varphi(t)}(f(x_0)-f(x^\star)) + \dfrac{h\,d\,L\,\sigma_*^2}{2}\displaystyle\int_0^t \dfrac{\psi(s)^2}{b(s)}e^{-2\mu(\varphi(t)-\varphi(s))}ds$ | 3 |
| **(H-)**,**(HRSI)** | $\left(\dfrac{1+2hL^2\mathfrak{T}}{\mathfrak{T}(\mu-2hL^2)}\right)^j \|x_0-x^*\|^2$ (with variance reduction) | 4 |

| Cond. | Rate (Discrete-time, no Gaussian assumption) | Thm. |
|-------|----------------------------------------------|------|
| $(\sim)$,**(H-)**,**(H$\sigma$)** | $\dfrac{2\,(f(x_0)-f(x^\star))}{(h\varphi_{k+1})} + \dfrac{h\,d\,L\,\sigma_*^2}{(h\varphi_{k+1})}\displaystyle\sum_{i=0}^{k}\dfrac{\psi_i^2}{b_i}h$ | E.1 |
| $(\sim)$,**(H-)**,**(H$\sigma$)**,**(HWQC)** | $\dfrac{\|x_0-x^\star\|^2}{\tau\,(h\varphi_{k+1})} + \dfrac{d\,h\,\sigma_*^2}{\tau\,(h\varphi_{k+1})}\displaystyle\sum_{i=0}^{k}\dfrac{\psi_i^2}{b_i}h$ | E.2 |
| **(H-)**,**(H$\sigma$)**,**(HWQC)** | $\dfrac{\|x_0-x^\star\|^2}{2\,\tau\,(h\varphi_{k+1})} + \dfrac{h\,d\,\sigma_*^2}{2\,\tau\,(h\varphi_{k+1})}\displaystyle\sum_{i=0}^{k}(1+\tau\varphi_{i+1}L)\dfrac{\psi_i^2}{b_i}h$ | E.2 |
| **(H-)**,**(H$\sigma$)**,**(HPŁ)** | $\displaystyle\prod_{i=0}^{k}(1-\mu\,h\psi_i)(f(x_0)-f(x^\star)) + \dfrac{h\,d\,L\,\sigma_*^2}{2}\sum_{i=0}^{k}\dfrac{\prod_{\ell=0}^{k}(1-\mu\,h\psi_\ell)}{\prod_{j=0}^{i}(1-\mu\,h\psi_l)}\dfrac{\psi_i^2}{b_i}h$ | E.3 |
| **(H-)**,**(HRSI)** | $\left(\dfrac{1+2L^2h^2m}{hm(\mu-2L^2h)}\right)^j \|x_0-x^*\|^2$ (with variance reduction) | E.4 |

Table 3: Summary of the main convergence results for MB-PGF and VR-PGF compared to SGD with mini-batch or variance reduced gradient estimators. $(\sim)$ indicates a randomized output. For the definition of the quantities in the rates, check App. D and App. E.

### A.1  Correspondences between continuous and discrete-time

We note the following simple correspondences:

1. $h$ corresponds to $dt$. The rates are *not* simplified to highlight the equivalence.

2. $h\varphi_{k+1}$ corresponds to $\varphi(t)$. Indeed, $\varphi(t) = \int_0^t \psi(s)ds \simeq \sum_{i=0}^{k}\psi_k h = \varphi_{k+1}h$.

3. The same argument holds for the exponential and the power, since $e^{at} \simeq (1+ah)^k$.

4. The rates for variance reduction match since by definition $\mathfrak{T} = m\,h$.

5. The difference only comes into a few constants which do not depend on the parameters of the problem nor on the algorithm. Those differences are due to higher order terms in the algorithm.

## A.2 Algebraic equivalence

In this section we motivate the equivalence outlined in Tb. 3 in the deterministic setting, although a similar derivation can easily be performed in the stochastic setting using the diffusion operator instead of the derivative (we introduce this object in App. B). We take inspiration from a concept in abstract algebra and we combine it with smoothness — a common assumption in optimization.

**Definition 1.** *Let $A$ be an algebra over a field $F$. A derivation is a linear map $D : A \to A$ that satisfies Leibniz's law: $D(ab) = aD(b) + D(a)b$.*

Consider the vector space of $d$-dimensional sequences over $\mathbb{N}$ equipped with pointwise and element-wise product and sum, which we denote as $\mathbb{R}^{d \times \infty}$; this is trivially an algebra. Next, let us define the sequence $D_h(x)$ (still in the algebra) pointwise: for all $k \in \mathbb{N}$

$$[D_h(x)]_k =: D_h(x, k) = \frac{x_{k+1} - x_k}{h}.$$

Notice that GD can be written as $D_h(x, k) = -\nabla f(x_k)$, which resembles the gradient flow equation $\frac{d}{dt} X(t) = -\nabla f(X(t))$. The *crucial question* is whether the continuous time derivative $\frac{d}{dt}$ and the operator $D_h$ have the same properties. This would motivate an algebraic equivalence between continuous and discrete time in optimization.

To start, we show that $D_h$ is *almost* a derivation. We denote by $x^+$ the one-step-ahead $x$ sequence: $x_k^+ = x_{k+1}$ for all $k \in \mathbb{N}$.

1. Let $x, y \in \mathbb{R}^{d \times \infty}$ and $k \in \mathbb{N}$, $D_h(x + y, k) = D_h(x, k) + D_h(y, k)$.

2. Let $x \in \mathbb{R}^{d \times \infty}$, $a \in \mathbb{R}$ and $k \in \mathbb{N}$, $D_h(ax, k) = aD_h(x, k)$.

3. Let $x, y \in \mathbb{R}^{d \times \infty}$; for all $k \in \mathbb{N}$,

$$D_h(xy, k) = \frac{1}{h}(y_{k+1} x_{k+1} - y_k x_k)$$

$$= \frac{1}{h}((y_{k+1} - y_k)x_{k+1} + y_k x_{k+1} - y_k x_k) = \frac{y_{k+1} - y_k}{h} x_{k+1} + y_k \frac{x_{k+1} - x_k}{h}.$$

Therefore $D(xy) = x^+ D_h(y) + D_h(x)y$.

Since we will only care about the value of $D_h(x)$ at iteration $k$, we are going to deal with the pointwise map $D_h(x, k)$ and deviate from the algebraic definition.

For a complete correspondence to continuous time, we still need a chain rule. For this, we need a bit more flexibility in the definition of $D_h$: let $g : \mathbb{R}^d \to \mathbb{R}$ be $L$-smooth, we define

$$D_h(g \circ x, k) := \frac{g(x_{k+1}) - g(x_k)}{h}.$$

Smoothness gives us a chain rule: we have

$$g(x_{k+1}) \leq g(x_k) + \langle \nabla g(x_k), x_{k+1} - x_k \rangle + \frac{L}{2}\|x_{k+1} - x_k\|^2;$$

hence

$$D_h(g \circ x, k) \leq \langle \nabla g(x_k), \frac{x_{k+1} - x_k}{h} \rangle + \frac{L}{2h}\|x_{k+1} - x_k\|^2 = \langle \nabla g(x_k), D_h(x, k) \rangle + \frac{Lh}{2}\|D_h(x, k)\|^2.$$

We condense our findings in the box below

> Let $\{x_k\}_{k \in \mathbb{N}}$ and $\{y_k\}_{k \in \mathbb{N}}$ be sequences of $\mathbb{R}^d$ vectors and let $g : \mathbb{R}^d \to \mathbb{R}$ be $L$-smooth.
> - Linearity : $D_h(x + y, k) = D_h(x, k) + D_h(y, k)$, $a \in \mathbb{R}$ and $D_h(ax, k) = aD_h(x, k)$.
> - Product rule: $D_h(xy, k) = D_h(x, k)y_k + x_{k+1}D_h(y, k)$.
> - Chain rule: $D_h(g(x), k) \leq \langle \nabla g(x_k), D_h(x, k) \rangle + \frac{Lh}{2}\|D_h(x, k)\|^2$.

This shows that the operations in continuous time and in discrete time are algebraically very similar, motivating the success behind the matching rates summarized in Tb. 3. Indeed, taking $h \to 0$ we recover the normal derivation rules from calculus.

# B Stochastic Calculus

In this appendix we summarize some important results in the analysis of Stochastic Differential equations [41, 46]. The notation and the results in this section will be used extensively in all proofs in this paper. We assume the reader to have some familiarity with Brownian motion and with the definition of stochastic integral (Ch. 1.4 and 1.5 in [41]).

## B.1 Itô's lemma and Dynkin's formula

We start with some notation: let $(\Omega, \mathcal{F}, \{\mathcal{F}(t)\}_{t \geq 0}, \mathbb{P})$ be a filtered probability space. We say that an event $E \in \mathcal{F}$ holds almost surely (a.s.) in this space if $\mathbb{P}(E) = 1$. We call $\mathcal{L}^p([a, b], \mathbb{R}^d)$, with $p > 0$, the family of $\mathbb{R}^d$-valued $\mathcal{F}(t)$-adapted processes $\{f(t)\}_{a \leq t \leq b}$ such that

$$\int_a^b \|f(t)\|^p dt \leq \infty.$$

Moreover, we denote by $\mathcal{M}^p([a, b], \mathbb{R}^d)$, with $p > 0$, the family of $\mathbb{R}^d$-valued processes $\{f(t)\}_{a \leq t \leq b}$ in $\mathcal{L}([a, b], \mathbb{R}^d)$ such that $\mathbb{E}\left[\int_a^b \|f(t)\|^p dt\right] \leq \infty$. We will write $h \in \mathcal{L}^p(\mathbb{R}_+, \mathbb{R}^d)$, with $p > 0$, if $h \in \mathcal{L}^p([0, T], \mathbb{R}^d)$ for every $T > 0$. Same definitions holds for matrix valued functions using the Frobenius norm $\|A\| := \sqrt{\sum_{ij} |A_{ij}|^2}$.

Let $B = \{B(t)\}_{t \geq 0}$ be a one dimensional Brownian motion defined on our probability space and let $X = \{X(t)\}_{t \geq 0}$ be an $\mathcal{F}(t)$-adapted process taking values on $\mathbb{R}^d$.

**Definition 2.** *Let $b \in \mathcal{L}^1(\mathbb{R}_+, \mathbb{R}^d)$ (the drift) and $\sigma \in \mathcal{L}^2(\mathbb{R}_+, \mathbb{R}^{d \times m})$ (the volatility). $X$ is an Itô process if it takes the form*

$$X(t) = x_0 + \int_0^t f(s)ds + \int_0^t \sigma(s)dB(s).$$

*We shall say that $X$ has the stochastic differential*

$$dX(t) = f(t)dt + \sigma(t)dB(t). \tag{3}$$

In this paper we indicate as $\partial_x f(x, t)$ the $d$-dimensional vector of partial derivatives of a scalar function $f : \mathbb{R}^d \times [0, \infty) \to \mathbb{R}$ w.r.t. each component of $x$. Moreover, we call $\partial_{xx} f(x, t)$ the $d \times d$-matrix of partial derivatives of each component of $\partial_x f(x, t)$ w.r.t each component of $x$. We now state the celebrated **Itô's lemma**.

---

**Theorem B.1** (Itô's lemma). *Let $X$ be an Itô process with stochastic differential $dX(t) = f(t)dt + \sigma(t)dB(t)$. Let $\mathcal{E}(x, t)$ be twice continuously differentiable in $x$ and continuously differentiable in t, taking values in $\mathbb{R}$. Then $\mathcal{E}(X(t), t)$ is again an Itô process with stochastic differential*

$$d\mathcal{E}(X(t), t) = \partial_t \mathcal{E}(X(t), t))dt + \langle \partial_x \mathcal{E}(X(t), t), f(t)\rangle dt$$
$$+ \frac{1}{2} \operatorname{Tr}\left(\sigma(t)\sigma(t)^T \partial_{xx}\mathcal{E}(X(t), t)\right) dt + \langle \partial_x \mathcal{E}(x(t), t), \sigma(t)\rangle dB(t), \quad (4)$$

*which we sometimes write as*

$$d\mathcal{E} = \partial_t \mathcal{E} dt + \langle \partial_x \mathcal{E}, dX\rangle + \frac{1}{2}\operatorname{Tr}\left(\sigma\sigma^T \partial_{xx}\mathcal{E}\right) dt$$

---

Following [41], we introduce the **Itô diffusion differential operator** $\mathscr{A}$:

$$\mathscr{A}(\cdot) = \partial_t(\cdot) + \langle \partial_x(\cdot), b(t)\rangle + \frac{1}{2}\operatorname{Tr}\left(\sigma(t)\sigma(t)^T \partial_{xx}(\cdot)\right). \tag{5}$$

It is then clear that, thanks to Itô's lemma,

$$d\mathcal{E}(X(t), t) = \mathscr{A}\mathcal{E}(X(t), t)dt + \langle \mathcal{E}_X(X(t), t), \sigma(t)dB(t)\rangle.$$

Moreover, by the definition of an Itô process, we know that at any time $t > 0$,

$$\mathcal{E}(X(t), t) = \mathcal{E}(x_0, 0) + \int_0^t \mathscr{A}\mathcal{E}(X(s), s)ds + \int_0^t \langle \partial_x \mathcal{E}(X(s), s), \sigma(s)dB(s) \rangle. \qquad a.s.$$

Taking the expectation the stochastic integral vanishes [16] and we have

$$\mathbb{E}[\mathcal{E}(X(t), t)] - \mathcal{E}(x_0, 0) = \mathbb{E}\left[ \int_0^t \mathscr{A}\mathcal{E}(X(t), t)dt \right]. \qquad (6)$$

This result can be generalized for stopping times and is known as **Dynkin's formula**.

## B.2 Stochastic differential equations

Stochastic Differential Equations (SDEs) are equations of the form

$$dX = b(X, t)dt + \sigma(X, t)dB(t).$$

Notice that this equation is different from Eq. (3), since $X$ also appears on the RHS. Hence, we need to define what it means for a stochastic process $X = \{X(t)\}_{t \geq 0}$ with values in $\mathbb{R}^d$ to solve an SDE.

**Definition 3.** *Let $X$ be as above with deterministic initial condition $X(0) = x_0$. Assume $b : \mathbb{R}^d \times [0, T] \to \mathbb{R}^d$ and $\sigma : \mathbb{R}^d \times [0, T] \to \mathbb{R}^{d \times m}$ are Borel measurable; $X$ is called a solution to the corresponding SDE if*

1. *$X$ is continuous and $\mathcal{F}(t)$-adapted;*

2. *$b \in \mathcal{L}^1 \left( [0, T], \mathbb{R}^d \right)$;*

3. *$\sigma \in \mathcal{L}^2 \left( [0, T], \mathbb{R}^{d \times m} \right)$;*

4. *For every $t \in [0, T]$*

$$X(t) = x_0 + \int_0^t b(X(s), s)ds + \int_0^t \sigma(X(s), s)dB(s) \quad a.s.$$

*Moreover, the solution $X(t)$ is said to be unique if any other solution $X^\star(t)$ is such that*

$$\mathbb{P}\left\{ X(t) = X^\star(t), \text{ for all } 0 \leq t \leq T \right\} = 1.$$

Notice that the solution to a SDE is an Itô process; hence we can use Itô's Formula (Thm. B.1). The following theorem gives a sufficient condition on $b$ and $\sigma$ for the existence of a solution to the corresponding SDE.

---

**Theorem B.2.** *Assume that there exist two positive constants $\bar{K}$ and $K$ such that*

1. *(Global Lipschitz condition) for all $x, y \in \mathbb{R}^d$ and $t \in [0, T]$*

$$\max\{\|b(x, t) - b(y, t)\|, \|\sigma(x, t) - \sigma(y, t)\|\} \leq \bar{K}\|x - y\|^2;$$

2. *(Linear growth condition) for all $x \in \mathbb{R}^d$ and $t \in [0, T]$*

$$\max\{\|b(x, t)\|, \|\sigma(x, t)\|\} \leq K(1 + \|x\|).$$

*Then, there exists a unique solution $X$ to the corresponding SDE , and $X \in \mathcal{M}^2([0, T], \mathbb{R}^d)$.*

---

**Numerical approximation.**   Often, SDEs are solved numerically. The simplest algorithm to provide a sample path $(\hat{x}_k)_{k \geq 0}$ for $X$, so that $X(k\Delta t) \cong x_k$ for some small $\Delta t$ and for all $k\Delta t \leq M$, is called Euler-Maruyama (Algorithm 1). For more details on this integration method and its approximation properties, the reader can check [41].

**Algorithm 1** Euler-Maruyama integration method for a SDE

---

**input** The drift $b$ and the volatility $\sigma$; the initial condition $x_0$
    fix a stepsize $\Delta t$;
    initialize $\hat{x}_0 = x_0$;
    $k = 0$;
    **while** $k \leq \lfloor \frac{T}{\Delta t} \rfloor$ **do**
        sample some $d$-dimensional Gaussian noise $Z_k \sim \mathcal{N}(0, I_d)$;
        compute $\hat{x}_{k+1} = \hat{x}_k + \Delta t\, b(\hat{x}_k, k\Delta t) + \sqrt{\Delta t}\, \sigma(\hat{x}_k, k\Delta t)Z_k$;
        $k = k + 1$;
    **end while**
**output** the approximated sample path $(\hat{x}_k)_{0 \leq k \leq \lfloor \frac{T}{\Delta t} \rfloor}$

---

### B.3 Functional SDEs

SDEs describe Markovian (also know as memoryless) processes: a Markovian process is a system where the current state completely determines the future evolution. Indeed, in an SDE, the RHS only depends on $X(t)$ and on $t$. To model variance-reduction methods such as SVRG [32], we will need a continuous time model which also retains some information about the past. This was also noted in [27].

First, we introduce Functional Stochastic Differential Equations (FSDEs) which are equations of the form
$$dX = b(X_{(0,t]}, t)dt + \sigma(X_{(0,t]}, t)dB(t),$$
where $X_{(0,t]}$ is the past history of $X$ up to time $t$. Here we focus on a particular type of FSDE, namely Stochastic Differential Delay Equations (SDDEs):

$$dX(t) = b(X(t), X(t - \xi(t)), t)dt + \sigma(X(t), X(t - \xi(t)), t)dB(t),$$

where $\xi(t) \in [0, \tau]$ is the delay at time $t$. As we did in the last subsection for SDEs, we need to define what it means for a stochastic process $X = \{X(t)\}_{t \geq -\tau}$ with values in $\mathbb{R}^d$ to solve an SDDE

**Definition 4.** *Let $X$ be as above with deterministic initial condition $X(s) = x_0$ for $-\tau \leq s \leq 0$. Assume $b : \mathbb{R}^d \times \mathbb{R}^d \times [0, T] \to \mathbb{R}^d$, $\xi : \mathbb{R}_+ \to [0, \tau]$ and $\sigma : \mathbb{R}^d \times \mathbb{R}^d \times [0, T] \to \mathbb{R}^{d \times m}$ are Borel measurable; $X$ is called a solution to the corresponding SDDE if*

1. *$X$ is continuous and $\mathcal{F}(t)$-adapted;*

2. *$b \in \mathcal{L}^1\left([0, T], \mathbb{R}^d\right)$;*

3. *$\sigma \in \mathcal{L}^2\left([0, T], \mathbb{R}^{d \times m}\right)$;*

4. *For every $t \in [0, T]$*

$$X(t) = x_0 + \int_0^t b(X(s), X(s - \xi(s)), s)ds + \int_0^t \sigma(X(s), X(s - \xi(s)), s)dB(s) \quad a.s.$$

*Moreover, a solution $X(t)$ is said to be unique if any other solution $X^\star(t)$ is such that*

$$\mathbb{P}\left\{X(t) = X^\star(t), \text{ for all } -\tau \leq t \leq T\right\} = 1.$$

We state now one existence and uniqueness theorem for SDDEs, which is adapted from equations 5.2 and 5.3 in [41].

**Theorem B.3.** *Assume that there exist two positive constants $\bar{K}$ and $K$ such that for all $x, \bar{x}, y, \bar{y} \in \mathbb{R}^d$ and for all $t \in [0, T]$*

    *1. (Lipschitz condition)*

$$\max\{\|b(x, y, t) - b(\bar{x}, \bar{y}, t)\|, \|\sigma(x, y, t) - \sigma(\bar{x}, \bar{y}, t)\|\} \leq \bar{K}(\|x - \bar{x}\| + \|y - \bar{y}\|);$$

    *2. (Linear growth condition)*

$$\max\{\|b(x, y, t)\|, \|\sigma(x, y, t)\|\} \leq K(1 + \|x\| + \|y\|).$$

*Then there exists a unique solution $X$ to the corresponding SDDE and $X \in \mathcal{M}^2([-\tau, T], \mathbb{R}^d)$.*

**Numerical approximation.** Often, SDSEs are solved numerically. Algorithm 1 can easily be modified to work with SDDEs (see (Algorithm 2)). For more details on approximation error SDDEs, we refer the reader to Chapter 5 in [41].

---

**Algorithm 2** Euler-Maruyama integration method for a SDDE

---

**input** The drift $b$ and the volatility $\sigma$; the initial condition $x_0$
    fix a stepsize $\Delta t$
    compute $q = \lfloor \frac{\tau}{\Delta t} \rfloor$;
    initialize $\hat{x}_k = x_0$ for $-q \leq k \leq 0$;
    $k = 0$;
    **while** $k \leq \lfloor \frac{T}{\Delta t} \rfloor$ **do**
        sample some $d$-dimensional Gaussian noise $Z_k \sim \mathcal{N}(0, I_d)$;
        compute $\hat{x}_{k+1} = \hat{x}_k + \Delta t\, b(\hat{x}_k, \hat{x}_{k-q}, k\Delta t) + \sqrt{\Delta t}\, \sigma(\hat{x}_k, \hat{x}_{k-q}, k\Delta t) Z_k$;
        $k = k + 1$;
    **end while**
**output** the approximated sample path $(\hat{x}_k)_{-q \leq k \leq \lfloor \frac{T}{\Delta t} \rfloor}$

---

## B.4 Time change in stochastic analysis

We conclude this appendix with a useful formula from [47], which is the equivalent to a chain rule for stochastic processes. We use this formula in Sec. 4.1.

**Theorem B.4** (Time change formula for Itô integrals)**.** *Let $c : \mathbb{R}_+ \to \mathbb{R}_+$ be a strictly positive continuous function and $\beta(t) = \int_0^t c(s)ds$. Denote by $\alpha(\cdot)$ the inverse of $\beta(\cdot)$ and suppose it is continuous. Let $\{B(t)\}_{t \geq 0}$ be an $m$-dimensional Brownian Motion and let the stochastic process $\{v(s)\}_{s \geq 0}$ with $v(s) \in \mathbb{R}^{n \times m}$ be Borel measurable in time, adapted to the natural filtration of $B$ and $\mathcal{M}^2(\mathbb{R}_+, \mathbb{R}^d)$. Define*

$$\tilde{B}(t) = \int_0^t \sqrt{c(s)} dB(s).$$

*Then $\{\tilde{B}(t)\}_{t \geq 0}$ is a Brownian Motion and we have*

$$\int_0^{\alpha(t)} v(s)dB(s) = \int_0^t \sqrt{\alpha'(s)} v(\alpha(s)) d\tilde{B}(s), \quad a.s.$$

## C Existence and Uniqueness of the solution of MB-PGF and VR-PGF

Let $A$ be a positive semidefinite matrix; by the spectral theorem, $A$ can be diagonalized as $A = VDV^T$, with $V$ an orthogonal matrix and $D$ a diagonal matrix with non-negative diagonal elements (the eigenvalues of $A$). We can define the principal square root $A^{1/2} := VD^{1/2}V^T$, where $D^{1/2}$ is the elementwise square root of $D$. It is clear that $A^{1/2}$ is also positive semidefinite and $A = A^{1/2}A^{1/2}$.

In this paper we analyze MB-PGF and VR-PGF, which we report below (see discussion and derivation in Sec. 2).

$$dX(t) = -\psi(t)\nabla f(X(t))\, dt + \psi(t)\sqrt{h/b(t)}\,\sigma_{\mathrm{MB}}(X(t))\, dB(t) \qquad \text{(MB-PGF)}$$

$$dX(t) = -\psi(t)\nabla f(X(t))\, dt + \psi(t)\sqrt{h/b(t)}\,\sigma_{\mathrm{VR}}(X(t), X(t-\xi(t)))\, dB(t) \qquad \text{(VR-PGF)}$$

The volatility of MB-SDE is defined as

$$\sigma_{\mathrm{MB}}(x) = \left( \frac{1}{N}\sum_{i=1}^{N} (\nabla f(x) - \nabla f_i(x))(\nabla f(x) - \nabla f_i(x))^T \right)^{1/2},$$

and a similar formula holds for the $\sigma_{\mathrm{VR}}(\cdot)$. From Thm. B.2 and Thm. B.3, we know that existence and uniqueness of the solution to the equations above requires this matrix valued function of $x$ to be Lipschitz continuous. Previous literature [52, 51, 42, 34], put this condition as a requirement at the beginning of their analysis. However, *since in our case we want to draw a direct connection to the algorithm*, we shall prove that Lipschitzianity is indeed verified.

To start, we remind again to the reader that in this paper we indicate as $\mathcal{C}_b^r(\mathbb{R}^d, \mathbb{R}^m)$ the family of $r$ times continuously differentiable functions from $\mathbb{R}^d$ to $\mathbb{R}^m$, with bounded $r$-th derivative. If $b$ is omitted, it means we just require $f$ to be $r$ times continuously differentiable.

A crucial lemma which can be found as Prop. 6.2 in [29] or as Thm. 5.2.3 in [59].

> **Lemma 1.** *Let $\Sigma : \mathbb{R}^n \to \mathbb{R}^{n\times n}$ be a $n \times n$ real positive semidefinite matrix function of an input vector $x \in \mathbb{R}^n$. Assume each component $\Sigma_{ij} : \mathbb{R}^n \to \mathbb{R}$ be in $\mathcal{C}_b^2(\mathbb{R}^n, \mathbb{R})$. Then, $\Sigma(x)^{1/2}$ is globally Lipschitz w.r.t. the Frobenius norm, meaning that there exists a constant $K$ such that for every $q, p \in \mathbb{R}^n$*
> $$\left\| \Sigma(q)^{1/2} - \Sigma(p)^{1/2} \right\| \le K\|q - p\|.$$

We proceed with the proofs of existence and uniqueness, which require the following assumption:

**(H)** Each $f_i$ is in $\mathcal{C}_b^3(\mathbb{R}^d, \mathbb{R})$ and is $L$-smooth.

> **Theorem C.1** (Existence and Uniqueness for MB-PGF). *Assume **(H)**. For all initial conditions $X(0) = x_0 \in \mathbb{R}^d$, MB-PGF has a unique solution (in the sense of Defs. 3 in App. B) on $[0,T]$, for any $T < \infty$. Let the stochastic process $X = \{X(t)\}_{0\le t\le T}$ be such solution; almost all (i.e. with probability 1) realizations of $X$ are continuous functions and $\mathbb{E}\left[ \int_0^T \|X(t)\|^2 dt \right] < \infty$.*

*Proof.* We basically need to check the conditions of Thm. B.2. First, we notice that $\nabla f$ and $\sigma_{\mathrm{MB}}$ are both Borel measurable because they are continuous.

Drift : We now verify the Lipschitz condition for the drift term. For every $t \le 0$ we trivially have that, since $\psi(t) \le 1$ and $f$ is $L$-smooth,

$$\|\psi(t)\nabla f(x) - \psi(t)\nabla f(y)\| \le \|\nabla f(x) - \nabla f(y)\| \le L\|x - y\|.$$

Next, we verify the linear growth condition. For every $t \ge 0$, using the reverse triangle inequality and the fact that $\psi(t) \in (0,1]$ and $\psi(0) = 1$,

$$L\|x\| \ge \|\psi(t)\nabla f(x) - \psi(t)\nabla f(0_d)\| \ge (|\|\psi(t)\nabla f(x)\| - \|\nabla f(0_d)\||).$$

Thus, we have linear growth with constant $K := \max\{\|\nabla f(0)\|, L\}$: for every $t \in [0,T]$ and $x \in \mathbb{R}^d$,

$$\|\psi(t)\nabla f(x)\| \le K(1 + \|x\|).$$

Volatility : We need to verify the same conditions for the volatility matrix $\sigma_{\mathrm{MB}}$. Let us define $g_i(x) := \nabla f(x) - \nabla f_i(x)$. Using the definition of Frobenius norm, the linearity of $\mathbb{E}$, the cyclicity of the trace functional, and the fact that $\psi(t) \in (0,1]$ for all $t \ge 0$, we get

$$\|\psi(t)\sqrt{h/b(t)}\,\sigma_{\mathrm{MB}}(x)\|^2 = \psi(t)^2 \frac{h}{b(t)}\,\mathrm{Tr}\left(\mathbb{E}\left[g_i(x)g_i(x)^T\right]\right)$$
$$= \psi(t)^2 \frac{h}{b(t)}\mathbb{E}\left[\mathrm{Tr}(g_i(x)^T g_i(x))\right] = \psi(t)^2 \frac{h}{b(t)}\mathbb{E}\|g_i(x)\|^2.$$

Since $g_i(x)$ is $L$-Lipschitz, by the same argument used above for the drift term, we have $\|g_i(x)\|^2 \leq C(1 + \|x\|^2)$ for some $C > 0$ and all $i \in [N]$. Plugging this in, since $b(t) \geq 1$

$$\|\psi(t)\sqrt{h/b(t)}\,\sigma_{\mathrm{MB}}(x)\|^2 = \leq D(1 + \|x\|^2),$$

for some finite positive $D$. To conclude the proof of linear growth, we notice that for any $p \in \mathbb{R}$, $\sqrt{1 + p^2} \leq 2(1 + |p|)$. Thus for $B := 2D$, we have

$$\|\psi(t)\sqrt{h/b(t)}\,\sigma_{\mathrm{MB}}(x)\| \leq B(1 + \|x\|).$$

Last, the global Lipschitzianity of $\sigma_{\mathrm{MB}}$ follows directly from Lemma 1 using the fact that $f$ is $\mathcal{C}_b^3(\mathbb{R}^d, \mathbb{R})$ and each $f_i$ is $\mathcal{C}_b^3(\mathbb{R}^d, \mathbb{R})$, because then the gradients are of class $\mathcal{C}^2$ and $\sigma_{\mathrm{MB}}$ is a smooth function of these gradients. ∎

---

**Theorem C.2** (Existence and Uniqueness for VR-PGF). *Assume **(H)**. For any initial condition $x_0$, such that $X(s) = x_0$ for all $t \in [-\tau, 0]$, VR-PGF has a unique solution (in the sense of Def. 4 in App. B) on $[-\tau, T]$, for any $T < \infty$. Moreover, let $X = \{X(t)\}_{0 \leq t \leq T}$ be such solution; almost all realizations of $X$ are continuous functions and $\mathbb{E}\left[\int_0^T \|X(t)\|^2 dt\right] < \infty$.*

---

*Proof.* This time we need to check the conditions of Thm. B.3. The requirements on the drift term are satisfied, as already shown in the proof for MB-PGF, since there is no delay in the drift. To verify the conditions on $\sigma_{\mathrm{VR}} : \mathbb{R}^d \times \mathbb{R}^d \to \mathbb{R}^{d \times d}$ we proceed again as in the proof for MB-PGF, using Lemma 1 but this time on the joint vector $(x, \tilde{x}) \in \mathbb{R}^d \times \mathbb{R}^d$ ($n$ in the lemma is $2d$), using the norm subadditivity. ∎

## D Convergence proofs in continuous-time

Fon convenience of the reader, we report here again the equations we are about to analyze continuous-time models, which we analyse in this paper, are

$$dX(t) = -\psi(t)\nabla f(X(t))\,dt + \psi(t)\sqrt{h/b(t)}\,\sigma_{\mathrm{MB}}(X(t))\,dB(t) \qquad \text{(MB-PGF)}$$

$$dX(t) = -\psi(t)\nabla f(X(t))\,dt + \psi(t)\sqrt{h/b(t)}\,\sigma_{\mathrm{VR}}(X(t), X(t - \xi(t)))\,dB(t) \qquad \text{(VR-PGF)}$$

where

- $\xi : \mathbb{R}_+ \to [0, \mathfrak{T}]$, the *staleness function*, is s.t. $\xi(hk) = \xi_k$ for all $k \geq 0$;
- $\psi(\cdot) \in \mathcal{C}^1(\mathbb{R}_+, [0, 1])$, the *adjustment function*, is s.t. $\psi(hk) = \psi_k$ for all $k \geq 0$ and $\frac{d\psi(t)}{dt} \leq 0$;
- $b(\cdot) \in \mathcal{C}^1(\mathbb{R}_+, \mathbb{R}_+)$, the *mini-batch size function* is s.t. $b(hk) = b_k$ for all $k \geq 0$ and $b(t) \geq 1$;
- $\{B(t)\}_{t \geq 0}$ is a $d-$dimensional Brownian Motion on some filtered probability space.

For existence and uniqueness we need to assume the following:

**(H)** Each $f_i(\cdot)$ is in $\mathcal{C}^3$ with bounded third derivative and $L$-smooth.

We also recall some of assumptions introduced in the main paper.

**(H$_{\mathrm{WQC}}$)** $f(\cdot)$ is $\mathcal{C}^1$ and exists $\tau > 0$ and $x^\star$ s.t. $\langle \nabla f(x), x - x^\star \rangle \geq \tau(f(x) - f(x^\star))$ for all $x \in \mathbb{R}^d$.

**(H$_{\mathrm{PL}}$)** $f(\cdot)$ is $\mathcal{C}^1$ and there exists $\mu > 0$ s.t. $\|\nabla f(x)\|^2 \geq 2\mu(f(x) - f(x^\star))$ for all $x \in \mathbb{R}^d$.

**(H$_{\mathrm{RSI}}$)** $f(\cdot)$ is $\mathcal{C}^1$ and there exists $\mu > 0$ s.t. $\langle \nabla f(x), x - x^\star \rangle \geq \frac{\mu}{2}\|x - x^\star\|^2$ for all $x \in \mathbb{R}^d$.

## D.1 Supporting lemmas

The following bound on the spectral norm also be found in [42, 34]. We report the proof for completeness.

---

**Lemma 2.** *Consider two symmetric $d-$dimensional square matrices $P$ and $Q$. We have*

$$\text{Tr}(PQ) \le d \cdot \|P\|_S \cdot \|Q\|_S.$$

---

*Proof.* Let $P_j$ and $Q_j$ be the j-th row(column) of $P$ and $Q$, respectively.

$$\text{Tr}(PQ) = \sum_{j=1}^{d} P_j^T Q_j \le \sum_{j=1}^{d} \|P_j\| \cdot \|Q_j\| \le \sum_{j=1}^{d} \|P\|_S \cdot \|Q\|_S = d \cdot \|P\|_S \cdot \|Q\|_S,$$

where we first used the Cauchy-Schwarz inequality, and then the following inequality:

$$\|A\|_S = \sup_{\|z\| \le 1} \|Az\| \ge \|Ae_j\| = \|A_j\|,$$

where $e_j$ is the j-th vector of the canonical basis of $\mathbb{R}^d$. ∎

We use the previous lemma to derive another key result below.

---

**Lemma 3.** *Assume (H). For any volatility matrix $\sigma(\cdot)$ such that $\|\sigma\sigma^T\|_S$ is upper bounded by $\sigma_*^2$, we have*

$$\text{Tr}\left(\sigma\sigma^T\right) \le d\sigma_*^2, \qquad \text{Tr}\left(\sigma\sigma^T\nabla^2 f(x)\right) \le Ld\sigma_*^2.$$

---

*Proof.* We will just prove the first inequality, since the proof for the second is very similar.

$$\text{Tr}\left(\sigma\sigma^T\nabla^2 f(x)\right) \le d\|\nabla^2 f(x)\|_S\|\sigma\sigma^T\|_S \le Ld\sigma_*^2,$$

where in the equality we used the cyclicity of the trace, in the first inequality we used Lemma 2 and in the last inequality we used and smoothness. ∎

## D.2 Analysis of MB-PGF

We provide a non-asymptotic analysis and then derive asymptotic rates.

### D.2.1 Non-asymptotic rates

These rates for MB-PGF are sketched in Sec. 3. We define $\varphi(t) := \int_0^t \psi(s)ds$. As [42, 34], we introduce a bound on the volatility in order to use Lemma 3.

**(H$\sigma$)** $\sigma_*^2 := \sup_{x \in \mathbb{R}^d} \|\sigma_{\text{MB}}(x)\sigma_{\text{MB}}(x)^T\|_S < \infty$, where $\|\cdot\|_S$ denotes the spectral norm.

---

**Theorem D.1** (Restated Thm. 1). *Assume (H), (H$\sigma$). Let $t > 0$ and $\tilde{t} \in [0, t]$ be a random time point with distribution $\frac{\psi(\tilde{t})}{\varphi(t)}$ for $\tilde{t} \in [0, t]$ (and $0$ otherwise). The solution to* MB-PGF *is s.t.*

$$\mathbb{E}\left[\|\nabla f(X(\tilde{t}))\|^2\right] \le \frac{f(x_0) - f(x^\star)}{\varphi(t)} + \frac{L\,d\,\sigma_*^2 h}{2\,\varphi(t)} \int_0^t \frac{\psi(s)^2}{b(s)}ds.$$

---

*Proof.* Define the energy $\mathcal{E} \in \mathcal{C}^2(\mathbb{R}^d, \mathbb{R}_+)$ such that $\mathcal{E}(x) := f(x) - f(x^\star)$. First, we find a bound on the infinitesimal diffusion generator of the stochastic process $\{\mathcal{E}(X(t))\}_{t \ge 0}$, which generalizes the concept of derivative for stochastic systems and is formally defined in App. B.1.

$$\mathscr{A}\mathcal{E}(X(t)) = \frac{h\psi(t)^2}{2b(t)} \text{Tr}\left(\sigma_{\text{MB}}(X(t))\sigma_{\text{MB}}(X(t))^T \partial_{xx}\mathcal{E}(X(t))\right) + \langle \partial_x \mathcal{E}(X(t)), -\psi(t)\nabla f(X(t))\rangle$$

$$\le \frac{h\,L\,d\,\psi(t)^2}{2b(t)}\sigma_*^2 - \psi(t)\|\nabla f(X(t))\|^2,$$

where in the inequality we used Lemma 3.

Note that the definition of $\mathscr{A}\mathcal{E}(X(t))$ in Eq. (5) does not include the term $\langle \partial_x \mathcal{E}, \sigma(t) dB(t) \rangle$ that vanishes when taking the expectation of the stochastic integral in Eq. (6). Therefore, integrating the bound above yields

$$\mathbb{E}[\mathcal{E}(X(t),t)] - \mathcal{E}(x_0,0) \le \frac{hLd\sigma_*^2}{2} \int_0^t \frac{\psi(s)^2}{b(s)} ds - \mathbb{E}\left[\int_0^t \psi(s)\|\nabla f(X(s))\|^2 ds\right]. \quad (7)$$

Next, notice that, since $\int_0^t \frac{\psi(s)}{\varphi(t)} dt = 1$, the function $s \mapsto \frac{\psi(s)}{\varphi(t)}$ defines a probability distribution. Let $\tilde{t} \in [0,t]$ have such distribution; using the law of the unconscious statistician

$$\mathbb{E}[\|\nabla f(X(\tilde{t})\|^2] = \frac{1}{\varphi(t)} \int_0^t \psi(s)\|\nabla f(X(s))\|^2 ds.$$

This trick was also used in the original SVRG paper [32]. To conclude, we plug in the last formula into Eq. (7):

$$\mathbb{E}[\mathcal{E}(X(t),t)] - \mathcal{E}(x_0,0) \le \frac{hLd\sigma_*^2}{2} \int_0^t \frac{\psi(s)^2}{b(s)} ds - \varphi(t)\mathbb{E}\left[\|\nabla f(X(\tilde{t})\|^2\right].$$

The result follows after dividing both sides by $\varphi(t)$, which is always positive for $t > 0$. ∎

---

**Theorem D.2** (Restated Thm. 2). *Assume (H), (Hσ), (Hwqc). Let $\tilde{t}$ be as in Thm. 1. The solution to __MB-PGF__ is s.t.*

$$\mathbb{E}\left[f(X(\tilde{t})) - f(x^\star)\right] \le \frac{\|x_0 - x^\star\|^2}{2\,\tau\,\varphi(t)} + \frac{h\,d\,\sigma_*^2}{2\,\tau\,\varphi(t)} \int_0^t \frac{\psi(s)^2}{b(s)} ds \quad (\text{W1})$$

$$\mathbb{E}\left[(f(X(t)) - f(x^\star))\right] \le \frac{\|x_0 - x^\star\|^2}{2\,\tau\,\varphi(t)} + \frac{h\,d\,\sigma_*^2}{2\,\tau\,\varphi(t)} \int_0^t (L\,\tau\,\varphi(s) + 1)\frac{\psi(s)^2}{b(s)} ds. \quad (\text{W2})$$

---

*Proof.* We prove the two formulas separately.

First formula. Define the energy $\mathcal{E} \in \mathcal{C}^2(\mathbb{R}^d, \mathbb{R}_+)$ such that $\mathcal{E}(x) := \frac{1}{2}\|x - x^\star\|^2$. First, we find a bound on the infinitesimal diffusion generator of the stochastic process $\{\mathcal{E}(X(t))\}_{t \ge 0}$.

$$\mathscr{A}\mathcal{E}(X(t)) =$$
$$= \frac{h\,\psi(t)^2}{2b(t)} \text{Tr}\left(\sigma_{\text{MB}}(X(t))\sigma_{\text{MB}}(X(t))^T \partial_{xx}\mathcal{E}(X(t))\right) + \langle \partial_x\mathcal{E}(X(t)), -\psi(t)\nabla f(X(t)) \rangle$$
$$\le \frac{h\,d\,\psi(t)^2}{2b(t)}\sigma_*^2 - \psi(t)\langle \nabla f(X(t)), X(t) - x^\star \rangle$$
$$\le \frac{h\,d\,\psi(t)^2}{2b(t)}\sigma_*^2 - \tau\psi(t)(f(X(t)) - f(x^\star)),$$

where in the first inequality we used Lemma 3 and in the second inequality we used weak-quasi-convexity. Integrating this bound (see Eq. (6)), we get

$$\mathbb{E}[\mathcal{E}(X(t),t)] - \mathcal{E}(x_0,0) \le \frac{hd\sigma_*^2}{2} \int_0^t \frac{\psi(s)^2}{b(s)} ds - \tau\mathbb{E}\left[\int_0^t \psi(s)(f(X(s)) - f(x^\star))ds\right].$$

Proceeding again as in the proof of Thm. 1 (above), we get the desired result.

Second formula : Define the energy $\mathcal{E} \in \mathcal{C}^2(\mathbb{R}^d \times \mathbb{R}, \mathbb{R}_+)$ such that $\mathcal{E}(x,t) := \tau\varphi(t)(f(x) - f(x^\star)) + \frac{1}{2}\|x - x^\star\|^2$. First, we find a bound on the infinitesimal diffusion generator of the stochastic

process $\{\mathcal{E}(X(t), t)\}_{t \geq 0}$.

$$\mathscr{A}\mathcal{E}(X(t), t) =$$

$$= \partial_t \mathcal{E}(X(t), t) + \frac{h\,\psi(t)^2}{2b(t)} \operatorname{Tr}\left(\sigma_{\mathrm{MB}}(X(t))\sigma_{\mathrm{MB}}(X(t))^T \partial_{xx}\mathcal{E}(X(t), t)\right)$$

$$+ \langle \partial_x \mathcal{E}(X(t), t), -\psi(t)\nabla f(X(t))\rangle$$

$$\leq \tau\psi(t)(f(X(t)) - f(x^\star)) + \frac{h\,d\,\psi(t)^2}{2b(t)}(L\tau\varphi(t) + 1)\sigma_*^2$$

$$+ \langle \tau\varphi(t)\nabla f(X(t)) + X(t) - x^\star, -\psi(t)\nabla f(X(t))\rangle$$

$$\leq \tau\psi(t)(f(X(t)) - f(x^\star)) - \psi(t)\langle \nabla f(X(t)), X(t) - x^\star\rangle + \frac{h\,d\,\psi(t)^2}{2b(t)}(L\tau\varphi(t) + 1)\sigma_*^2$$

$$\leq \frac{h\,d\,\sigma_*^2}{2} \frac{(L\tau\varphi(t) + 1)\psi(t)^2}{b(t)},$$

where in the first inequality we used the fact that $\dot{\varphi}(t) = \psi(t)$ and Lemma 3; in the second inequality we discarded a negative term; in the third inequality we used weak-quasi-convexity. Next, after integration (see Dynkin formula Eq. (6)), plugging in the definition of $\mathcal{E}$, we get

$$\tau\varphi(t)\mathbb{E}\left[f(X(t)) - f(x^\star)\right] + \frac{1}{2}\mathbb{E}\left[\|X(t) - x^\star\|^2\right] \leq \frac{1}{2}\|x_0 - x^\star\|^2 + \frac{dh\sigma_*^2}{2}\int_0^t \frac{(L\tau\varphi(s) + 1)\psi(s)^2}{b(s)}.$$

Discarding the positive term $\frac{1}{2}\mathbb{E}\left[\|X(t) - x^\star\|^2\right]$ on the LHS and dividing[17] everything by $\tau\varphi(t)$ we get the result. ∎

---

**Theorem 6** (Restated Thm. 3). *Assume (H), (Hσ), (HPŁ). The solution to MB-PGF is s.t.*

$$\mathbb{E}[f(X(t)) - f(x^\star)] \leq e^{-2\mu\varphi(t)}(f(x_0) - f(x^\star)) + \frac{h\,L\,d\,\sigma_*^2}{2}\int_0^t \frac{\psi(s)^2}{b(s)}e^{-2\mu(\varphi(t) - \varphi(s))}ds.$$

---

*Proof.* Define the energy $\mathcal{E} \in \mathcal{C}^2(\mathbb{R}^d \times \mathbb{R}, \mathbb{R}_+)$ such that $\mathcal{E}(x, t) := e^{2\mu\varphi(t)}(f(x) - f(x^\star))$. First, we find a bound on the infinitesimal diffusion generator of the stochastic process $\{\mathcal{E}(X(t), t)\}_{t \geq 0}$.

$$\mathscr{A}\mathcal{E}(X(t), t) =$$

$$= \partial_t \mathcal{E}(X(t), t) + \frac{h\,\psi(t)^2}{2b(t)} \operatorname{Tr}\left(\sigma_{\mathrm{MB}}(X(t))\sigma_{\mathrm{MB}}(X(t))^T \partial_{xx}\mathcal{E}(X(t), t)\right)$$

$$+ \langle \partial_x \mathcal{E}(X(t), t), -\psi(t)\nabla f(X(t))\rangle$$

$$\leq 2\mu\,\psi(t)\,e^{2\mu\varphi(t)}(f(X(t)) - f(x^\star)) + \frac{h\,d\,L\,\psi(t)^2}{2b(t)}\sigma_*^2 e^{2\mu\varphi(t)} - \psi(t)\,e^{2\mu\varphi(t)}\|\nabla f(X(t))\|^2$$

$$\leq \frac{h\,d\,L\,\psi(t)^2}{2b(t)}\sigma_*^2 e^{2\mu\varphi(t)},$$

where in the first inequality we used the fact that $\dot{\varphi}(t) = \psi(t)$ and Lemma 3 and in the second inequality we used the PŁ assumption.

Finally, after integration (see Eq. (6)), plugging in the definition of $\mathcal{E}$, we get

$$e^{2\mu\varphi(t)}\mathbb{E}[f(X(t)) - f(x^\star)] \leq f(x_0) - f(x^\star) + \frac{h\,d\,L\,\sigma_*^2}{2}\int_0^t \frac{\psi(s)^2}{2b(s)}e^{2\mu\varphi(s)}ds.$$

The statement follows once we divide everything by $e^{2\mu\varphi(t)}$. ∎

### D.3 Asymptotic rates for decreasing adjustment function

> **Corollary 1.** *Assume (H), (Hσ). Let $t > 0$ and $\tilde{t} \in [0,t]$ be a random time point with distribution $\frac{\psi(\tilde{t})}{\varphi(t)}$ for $\tilde{t} \in [0,t]$ (and 0 otherwise). If $\psi(\cdot)$ has the form $\psi(t) = 1/(t+1)^a$ and $b(t) = b \geq 1$ then MB-PGF is s.t.*
>
> $$\mathbb{E}\left[\|\nabla f(X(\tilde{t}))\|^2\right] \leq \begin{cases} \mathcal{O}\left(\frac{1}{t^a}\right) & 0 < a < \frac{1}{2} \\ \mathcal{O}\left(\frac{\log(t)}{\sqrt{t}}\right) & a = \frac{1}{2} \\ \mathcal{O}\left(\frac{1}{t^{1-a}}\right) & \frac{1}{2} < a < 1 \\ \mathcal{O}\left(\frac{1}{\log(t)}\right) & a = 1 \end{cases}.$$

*Proof.* Thanks to Prop. 1, we have

$$\mathbb{E}\left[\|\nabla f(X(\tilde{t}))\|^2\right] \leq \frac{f(x_0) - f(x^\star)}{\varphi(t)} + \frac{L\, d\, \sigma_*^2 h}{2\, b\, \varphi(t)} \int_0^t \psi(s)^2 ds.$$

First, notice that if $a > 1$, $\lim_{t\to\infty} \varphi(t) < \infty$ and we cannot retrieve convergence. Else, for $0 < a < 1$, the deterministic term $\frac{f(x_0)-f(x^\star)}{\varphi(t)}$ is $\mathcal{O}\left(t^{1-a}\right)$ and $\mathcal{O}\left(\log^{-1}(t)\right)$ for $a = 1$. The stochastic term $\frac{1}{\varphi(t)} \int_0^t \psi(s)^2 ds$ is $\mathcal{O}\left(t^{-a}\right)$ for $a \in (0,1/2) \cup (1/2,1)$, $\mathcal{O}\left(\frac{\log(t)}{\sqrt{t}}\right)$ for $a = \frac{1}{2}$ and $\mathcal{O}(1)$ for $a = 1$. The assertion follows combining asymptotic rates just derived for the deterministic and the stochastic term. ∎

> **Corollary 2.** *Assume (H), (Hσ), (Hwqc). Let $\tilde{t}$ be as in Thm. 1. If $\psi(\cdot)$ has the form $\psi(t) = 1/(t+1)^a$ and $b(t) = b \geq 1$, then the solution to MB-PGF is s.t.*
>
> $$\mathbb{E}\left[f(X(\tilde{t})) - f(x^\star)\right] \leq \begin{cases} \mathcal{O}\left(\frac{1}{t^a}\right) & 0 < a < \frac{1}{2} \\ \mathcal{O}\left(\frac{\log(t)}{\sqrt{t}}\right) & a = \frac{1}{2} \\ \mathcal{O}\left(\frac{1}{t^{1-a}}\right) & \frac{1}{2} < a < 1 \\ \mathcal{O}\left(\frac{1}{\log(t)}\right) & a = 1 \end{cases}.$$
>
> *Moreover, for $\frac{1}{2} \leq a \leq 1$ we can avoid taking a randomized time point:*
>
> $$\mathbb{E}\left[f(X(t)) - f(x^\star)\right] \leq \begin{cases} \mathcal{O}\left(\frac{1}{t^{2a-1}}\right) & \frac{1}{2} < a < \frac{2}{3} \\ \mathcal{O}\left(\frac{\log(t)}{t^{1/3}}\right) & a = \frac{2}{3} \\ \mathcal{O}\left(\frac{1}{t^{1-a}}\right) & \frac{2}{3} < a < 1 \\ \mathcal{O}\left(\frac{1}{\log(t)}\right) & a = 1 \end{cases}.$$

*Proof.* The first part is identical to Cor. 1 using this time Prop. 2. Regarding the second part, again from Prop. 2 we have

$$\mathbb{E}\left[f(X(\tilde{t})) - f(x^\star)\right] \leq \frac{\|x_0 - x^\star\|^2}{2\, \tau\, \varphi(t)} + \frac{h\, d\, \sigma_*^2}{2\, \tau\, b\, \varphi(t)} \int_0^t \psi(s)^2 ds.$$

The deterministic term $\frac{1}{2\tau\varphi(t)}\|x_0 - x^\star\|^2$ is $\mathcal{O}\left(\frac{1}{t^{1-a}}\right)$ for $0 < a < 1$, $\mathcal{O}\left(\frac{1}{\log(t)}\right)$ for $a = 1$ and $\mathcal{O}(1)$ (i.e. does not converge to 0) for $a > 1$.

The stochastic term $\frac{h\, d\, \sigma_*^2}{2\, \tau\, \varphi(t)} \int_0^t \psi(s)^2 ds$ requires a more careful analysis : first of all notice that $(L\tau\varphi(t) + 1)\psi(t)^2$ is $\mathcal{O}\left(\max\left\{t^{1-3a}, t^{-2a}\right\}\right)$. Hence its integral is $\mathcal{O}\left(\max\left\{t^{2-3a}, t^{1-2a}\right\}\right)$ for $\frac{1}{2} < a < \frac{2}{3}$, is $\mathcal{O}(1)$ for $a > \frac{2}{3}$ and has a more complicated asymptotic behavior for $a \neq \frac{1}{2}, \frac{2}{3}$. First, it is clear that, since the integral is bounded for $\frac{2}{3} < a$, the asymptotic convergence rate in this case is

$\mathcal{O}\left(\frac{1}{\varphi(t)}\right) = \mathcal{O}\left(\frac{1}{t^{1-a}}\right)$ for $\frac{2}{3} < a < 1$ and $\mathcal{O}\left(\frac{1}{\log(t)}\right)$ for $a = 1$. Next, we get the two pathological cases out of our way:

- For $a = \frac{1}{2}$ we do not have converge, since the partial integral term $\frac{d\sigma_*^2}{2\tau\varphi(t)} \int_0^t L\tau\varphi(s)\psi(s)^2 ds$ is of the same order as $\varphi(t)$.

- For $a = \frac{2}{3}$, $\frac{d\sigma_*^2}{2\tau\varphi(t)} \int_0^t (L\tau\varphi(s) + 1)\psi(s)^2 ds$ is $\mathcal{O}\left(\log(t)\right)$. Hence, the resulting asymptotic bound is $\mathcal{O}\left(\frac{\log(t)}{\varphi(t)}\right) = \mathcal{O}\left(\frac{\log(t)}{t^{1/3}}\right)$.

Last, since for $\frac{1}{2} < a < \frac{2}{3}$ the integral term is $\mathcal{O}\left(\max\left\{t^{2-3a}, t^{1-2a}\right\}\right) = \mathcal{O}(t^{2-3a})$, the convergence rate is $\mathcal{O}\left(\frac{t^{2-3a}}{\varphi(t)}\right) = \mathcal{O}(t^{1-2a})$. This completes the proof of the assertion. ∎

**Remark 1.** *The best achievable rate in the context of the previous corollary is corresponding to $\psi(t) = \frac{1}{\sqrt{t}}$ if we look at the infimum, but is instead corresponding to $\psi(t) = \frac{1}{t^{2/3}}$ if we just look at the final point.*

---

**Corollary 3.** *Assume (H), (Hσ), (Hₚₗ). If $\psi(\cdot)$ has the form $\psi(t) = 1/(t+1)^a$ and $b(t) = b \geq 1$, then he solution to MB-PGF is s.t.*

$$\mathbb{E}\left[f(X(t)) - f(x^\star)\right] \leq \mathcal{O}\left(\frac{1}{t^a}\right).$$

---

*Proof.* We start from Prop. 3:

$$\mathbb{E}[f(X(t)) - f(x^\star)] \leq e^{-2\mu\varphi(t)}(f(x_0) - f(x^\star)) + \frac{h\,L\,d\,\sigma_*^2}{2b} \int_0^t \psi(s)^2 e^{-2\mu(\varphi(t)-\varphi(s))} ds.$$

For $0 < a < 1$, the term $e^{-2\mu\varphi(t)}$ goes down exponentially fast. Thus, we just need to consider the second addend. Let $\hat{t} \in [0, t]$, then

$$\int_0^t \psi(s)^2 e^{-2\mu(\varphi(t)-\varphi(s))} ds \leq \int_0^{\hat{t}} \psi(s)^2 e^{-2\mu(\varphi(t)-\varphi(s))} ds + \int_{\hat{t}}^t \psi(s)^2 e^{-2\mu(\varphi(t)-\varphi(s))} ds$$

$$\leq e^{-2\mu(\varphi(t)-\varphi(\hat{t}))} \int_0^{\hat{t}} \psi(s)^2 ds + \frac{\psi(\hat{t})}{2\mu} \int_{\hat{t}}^t 2\mu\psi(s) e^{-2\mu(\varphi(t)-\varphi(s))} ds.$$

Pick $\hat{t} = t/2$, notice that, since for $\psi(t) = \frac{1}{(1+t)^a}$, $\int_0^{t/2} \psi(s)^2 ds$ grows at most polynomially in $t$. Hence, first addend in the last formula decays exponentially fast. Then again we just need to consider the second addend of the last formula; in particular notice that

$$\int_{\hat{t}}^t 2\mu\psi(s) e^{-2\mu(\varphi(t)-\varphi(s))} ds = e^{-2\mu\varphi(t)} \int_{\hat{t}}^t 2\mu\psi(s) e^{2\mu\varphi(s)} ds$$

$$= e^{-2\mu\varphi(t)} \left(e^{2\mu\varphi(t)} - e^{2\mu\varphi(t/2)}\right)$$

$$= 1 - e^{-2\mu(\varphi(t)-\varphi(t/2))}.$$

Hence, for $t$ big enough, the considered integral will be less than 1. All in all, we asymptotically have $\mathbb{E}[f(X(t)) - f(x^\star)] \leq \mathcal{O}(\psi(t))$, which gives the desired result.

∎

**Remark 2.** *We retrieve in continuous time the bound in [44]: the rate is always $\Omega\left(\frac{1}{t}\right)$.*

Figure 3: Simulation of MB-PGF for $f(x) = \frac{1}{2}\mu\|x\|^2$ with $x \in \mathbb{R}^2$, $\sigma_*^2 = 0.1$ and $\mu = 2$. Simulation with Euler-Maruyama (stepsize = $10^{-4}$).

Figure 4: Simulation of MB-PGF for $f(x) = \frac{1}{2}\mu\|x\|^2$ with $x \in \mathbb{R}^{100}$, $\sigma_*^2 = 0.1$ and $\mu = 2$. Simulation with Euler-Maruyama (stepsize = $10^{-4}$).

| Condition | Limit | Bound |
|---|---|---|
| **(H)**, **(H$\sigma$)** | $\lim_{t\to\infty} \mathbb{E}\left[\|\nabla f(X(\tilde{t}))\|^2\right]$ | $\frac{Ld\sigma_*^2}{2b}$ |
| **(H)**, **(H$\sigma$)**, **(H$_{\text{WQC}}$)** | $\lim_{t\to\infty} \mathbb{E}\left[f(X(\tilde{t})) - f(x^\star)\right]$ | $\frac{Ld\sigma_*^2}{2\tau b}$ |
| **(H)**, **(H$\sigma$)**, **(H$_{\text{PŁ}}$)** | $\lim_{t\to\infty} \mathbb{E}\left[f(X(t)) - f(x^\star)\right]$ | $\frac{Ld\sigma_\#^2}{4\mu b}$ |

Table 4: Ball of convergence of MB-PGF under constant $\psi(t) = 1$, $b(t) = b$. For $t > 0$, $\tilde{t} \in [0, t]$ has probability distribution $\frac{\psi(s)}{\varphi(t)}$ for $s \in [0, t]$ (and 0 otherwise).

#### D.3.1 Limit sub-optimality under constant adjustment function

In this paragraph we pick $\psi(t) = 1$. The results can be found in Tb. 4. The only non-obvious limit is the one for PŁ functions. By direct calculation,

$$\mathbb{E}[f(X(t)) - f(x^\star)] \le e^{-2\mu\varphi(t)}(f(x_0) - f(x^\star)) + \frac{h\,d\,L}{2b}\int_0^t \sigma_*^2 e^{-2\mu(t-s)}ds$$

$$= e^{-2\mu\varphi(t)}(f(x_0) - f(x^\star)) + \frac{h\,d\,L\,\sigma_*^2}{2}\frac{1 - e^{-2\mu t}}{2\,b\,\mu}.$$

The result follows taking the limit.

**Example D.1.** *We can verify the results in Tb. 4 using the quadratic function $f(x) = \frac{\mu}{2}\|x\|^2$, which is PŁ. This function is isotropic, so $\mu = L$. Under persistent noise $\sigma_*^2 I_d$, where $I_d$ is the identity matrix, the MB-PGF is $dX(t) = -\mu X(t)dt + h\sigma_* dB(t)$. This has solution $\mathbb{E}[f(X(t)) = f(x_0)e^{-2\mu t} + \frac{hd\sigma_*^2}{4}$, which perfectly matches the bound in Tb. 4. In Fig. 3 and 4 one can see a simulation for $d = 1$ and $d = 100$, keeping the noise constant at $\sigma_*^2 = 0.1$ and $\mu = 2$. One can clearly see that the bound is increasing with the number of dimensions. Moreover, by the law of large numbers, the variance in $f(X)$ is decreasing with the number of dimensions (it is a sum of $\chi^2$ distributions).*

### D.4 Analysis of VR-PGF

We remind the reader that the SVRG gradient estimate (see Sec. 2), with mini-batch size $b(t) = 1$ (always assumed here) is defined as

$$\mathcal{G}_{\text{VR}}(x_k) := \nabla f_{i_k}(x_k) - \nabla f_{i_k}(\tilde{x}_k) + \nabla f(\tilde{x}_k),$$

where $f(x) = \frac{1}{N}\sum_{i=1}^{N} f_i(x)$, with $\{f_i\}_{i=1}^{N}$ a collection of functions s.t. $f_i : \mathbb{R}^d \to \mathbb{R}$ for any $i \in \{1, \cdots, N\}$. We call $x^\star$ the unique global minimum of $f$. The stochastic gradient index $i_k$ is sampled uniformly from $\{1, \ldots, N\}$ and $\tilde{x}_k \in \{x_0, x_1, \ldots, x_{k-1}\}$ is the pivot used at iteration $k$. SVRG builds a sequence $(x_k)_{k\geq 0}$ of estimates of the solution $x^\star$ in a recursive way:

$$x_{k+1} = x_k - h\mathcal{G}_{\text{VR}}(x_k, \tilde{x}_{k-\xi_k}), \tag{SVRG}$$

where $h \geq 0$. $\xi_k$ is picked to be the sawtooth wave function with period $m \in \mathbb{N}_+$. Also, after $m$ iterations, the standard discrete-time SVRG analysis [32, 53, 54, 3, 4] requires **"jumping"** and set $x_k = x_{\hat{r}_k}$, where $\hat{r}_k$ is picked at random from $\{k - m, \ldots, k - 1\}$. This is known as *Option II* [32], as opposed to *Option I* which performs no jumps. The latter variant is widely used in practice [26], but, unfortunately, is not typically analyzed in the discrete-time literature.

As in App. E.1.1, we denote by $\{\mathcal{F}_k\}_{k\geq 0}$ the natural filtration induced by the stochastic process with jumps $\{x_k\}_{k\geq 0}$. The conditional mean and covariance matrix of $\mathcal{G}_{\text{VR}}$ are

$$\mathbb{E}_{\mathcal{F}_{k-1}}[\mathcal{G}_{\text{VR}}(x_k)] = \nabla f(x_k), \tag{8}$$

$$\Sigma_{\text{VR}}(x_k, \tilde{x}_k) := \mathbb{Cov}_{\mathcal{F}_{k-1}}[\mathcal{G}_{\text{VR}}(x_k)] \tag{9}$$

$$= \mathbb{E}_{\mathcal{F}_{k-1}}\left[(\mathcal{G}_{\text{VR}}(x_k) - \nabla f(x_k))(\mathcal{G}_{\text{VR}}(x_k) - \nabla f(x_k))^T\right].$$

We start with a lemma and a corollary, which will be used both in continuous and in discrete time and that are partially derived in [32] and [4].

---

**Lemma 4.** *Assume (H). We have*

$$\text{Tr}\left(\Sigma_{VR}(x_k, \tilde{x}_k)\right) \leq \mathbb{E}_{\mathcal{F}_{k-1}}\left[\|\mathcal{G}_{VR}(x_k)\|^2\right]$$
$$\leq 2\mathbb{E}_{\mathcal{F}_{k-1}}\|\nabla f_i(x_k) - \nabla f_i(x^\star)\|^2 + 2\mathbb{E}_{\mathcal{F}_{k-1}}\|\nabla f_i(\tilde{x}_k) - \nabla f_i(x^\star))\|^2.$$

---

*Proof.* Let us define $\epsilon_{\text{VR}}(x_k, \tilde{x}_k) := \mathcal{G}_{\text{VR}}(x_k, \tilde{x}_k) - \nabla f(x_k)$. First notice that, $\epsilon_{\text{VR}}(x_k, \tilde{x}_k)$ has zero mean, and

$$\text{Tr}\left(\Sigma_{\text{VR}}(x_k, \tilde{x}_k)\right) = \text{Tr}\left(\mathbb{E}_{\mathcal{F}_{k-1}}[\epsilon_{\text{VR}}(x_k, \tilde{x}_k)\epsilon_{\text{VR}}(x_k, \tilde{x}_k)^T]\right) =$$
$$= \mathbb{E}_{\mathcal{F}_{k-1}}\left[\text{Tr}(\epsilon_{\text{VR}}(x_k, \tilde{x}_k)\epsilon_{\text{VR}}(x_k, \tilde{x}_k)^T\right]$$
$$= \mathbb{E}_{\mathcal{F}_{k-1}}\left[\text{Tr}(\epsilon_{\text{VR}}(x_k, \tilde{x}_k)^T\epsilon_{\text{VR}}(x_k, \tilde{x}_k))\right] = \mathbb{E}_{\mathcal{F}_{k-1}}\|\epsilon_{\text{VR}}(x_k, \tilde{x}_k)\|^2, \quad (10)$$

where the second equality is given by the linearity of the trace and third equality by the cyclic property of the trace. Notice that, since for any random variable $\zeta$ we have $\mathbb{E}_{\mathcal{F}_{k-1}}[\|\zeta - \mathbb{E}_{\mathcal{F}_{k-1}}\zeta\|^2] = \mathbb{E}_{\mathcal{F}_{k-1}}[\|\zeta\|^2] - \|\mathbb{E}_{\mathcal{F}_{k-1}}[\zeta]\|^2 \leq \mathbb{E}_{\mathcal{F}_{k-1}}[\|\zeta\|^2]$, then

$$\mathbb{E}_{\mathcal{F}_{k-1}}[\|\epsilon_{\text{VR}}(x_k, \tilde{x}_k)\|^2] \leq \mathbb{E}_{\mathcal{F}_{k-1}}[\|\mathcal{G}_{\text{VR}}(x_k, \tilde{x}_k)\|^2].$$

Hence, we found that $\text{Tr}\left(\Sigma_{\text{VR}}(x_k, \tilde{x}_k)\right) \leq \mathbb{E}_{\mathcal{F}_{k-1}}[\|\mathcal{G}_{\text{VR}}(x)\|^2]$. We further bound this term with a simple calculation

$$\mathbb{E}_{\mathcal{F}_{k-1}}\|\mathcal{G}_{\text{VR}}(x)\|^2 = \mathbb{E}_{\mathcal{F}_{k-1}}\|\nabla f_i(x_k) - \nabla f_i(\tilde{x}_k) + \nabla f(\tilde{x}_k)\|^2$$
$$= \mathbb{E}_{\mathcal{F}_{k-1}}\|\nabla f_i(x_k) - \nabla f_i(x^\star) - [\nabla f_i(\tilde{x}_k) - \nabla f_i(x^\star) - \nabla f(\tilde{x}_k)]\|^2$$
$$\leq 2\mathbb{E}_{\mathcal{F}_{k-1}}\|\nabla f_i(x_k) - \nabla f_i(x^\star)\|^2 + 2\mathbb{E}\|\nabla f_i(\tilde{x}_k) - \nabla f_i(x^\star) - \nabla f(\tilde{x}_k)\|^2$$
$$= 2\mathbb{E}_{\mathcal{F}_{k-1}}\|\nabla f_i(x_k) - \nabla f_i(x^\star)\|^2$$
$$\quad + 2\mathbb{E}_{\mathcal{F}_{k-1}}\|\nabla f_i(\tilde{x}_k) - \nabla f_i(x^\star) - \mathbb{E}_{\mathcal{F}_{k-1}}[\nabla f(\tilde{x}_k) - \nabla f_i(x^\star)]\|^2$$
$$\leq 2\mathbb{E}_{\mathcal{F}_{k-1}}\|\nabla f_i(x_k) - \nabla f_i(x^\star)\|^2 + 2\mathbb{E}_{\mathcal{F}_{k-1}}\|\nabla f_i(\tilde{x}_k) - \nabla f_i(x^\star))\|^2, \tag{11}$$

where in the first inequality we used the parallelogram law; in the third equality we used $\mathbb{E}_{\mathcal{F}_{k-1}}[\nabla f_i(x^\star)] = 0$ and in the second inequality we used again the fact that for any random variable $\zeta$, $\mathbb{E}_{\mathcal{F}_{k-1}}\|\zeta - \mathbb{E}_{\mathcal{F}_{k-1}}\zeta\|^2 = \mathbb{E}_{\mathcal{F}_{k-1}}\|\zeta\|^2 - \|\mathbb{E}_{\mathcal{F}_{k-1}}\zeta\|^2 \leq \mathbb{E}_{\mathcal{F}_{k-1}}\|\zeta\|^2$. ∎

Using the previous lemma, we can derive the following result.

---

**Corollary 4.** *Assume (H). Then*

$$\text{Tr}\left(\Sigma_{VR}(x_k, \tilde{x}_k)\right) \leq \mathbb{E}_{\mathcal{F}_{k-1}}\left[\|\mathcal{G}_{VR}(x_k)\|^2\right]$$
$$\leq 2L^2\mathbb{E}_{\mathcal{F}_{k-1}}\left[\|x_k - x^\star\|^2\right] + 2L^2\mathbb{E}_{\mathcal{F}_{k-1}}\left[\|\tilde{x}_k - x^\star\|^2\right].$$

---

*Proof.* Using first smoothness we have, starting from Lemma 4

$$\text{Tr}\left(\Sigma_{VR}(x_k, \tilde{x}_k)\right) \leq \mathbb{E}_{\mathcal{F}_{k-1}}\left[\|\mathcal{G}_{VR}(x_k)\|^2\right]$$
$$\leq 2\mathbb{E}_{\mathcal{F}_{k-1}}\|\nabla f_i(x_k) - \nabla f_i(x^\star)\|^2 + 2\mathbb{E}_{\mathcal{F}_{k-1}}\|\nabla f_i(\tilde{x}_k) - \nabla f_i(x^\star))\|^2$$
$$\leq 2L^2\mathbb{E}_{\mathcal{F}_{k-1}}\left[\|x_k - x^\star\|^2\right] + 2L^2\mathbb{E}_{\mathcal{F}_{k-1}}\left[\|\tilde{x}_k - x^\star\|^2\right].$$

∎

Next, we provide a convergence rate for *Option II*.

### D.4.1 Convergence rate under Option II

We consider, the case $b(t) = \psi(t) = 1$. Therefore, VR-PGF reads

$$dX(t) = -\nabla f(X(t))\,dt + \sqrt{h}\,\sigma_{VR}(X(t), X(t - \xi(t)))\,dB(t).$$

As for standard SVRG with Option II, every $\mathfrak{T}$ seconds we perform a jump.

---

**Theorem 7** (Restated Thm. 4). *Assume (H), (H$_{RSI}$) and choose $\xi(t) = t - \sum_{j=1}^{\infty} \delta(t - j\mathfrak{T})$ (saw-tooth wave), where $\delta(\cdot)$ is the Dirac delta. Let $\{X(t)\}_{t \geq 0}$ be the solution to VR-PGF with additional jumps at times $(j\mathfrak{T})_{j \in \mathbb{N}}$: we pick $X(j\mathfrak{T} + \mathfrak{T})$ uniformly in $\{X(s)\}_{j\mathfrak{T} \leq s < (j+1)\mathfrak{T}}$. Then,*

$$\mathbb{E}[\|X(j\mathfrak{T}) - x^\star\|^2] = \left(\frac{2hL^2\mathfrak{T} + 1}{\mathfrak{T}(\mu - 2hL^2)}\right)^j \|x_0 - x^*\|^2.$$

---

*Proof.* Define the energy $\mathcal{E} \in \mathcal{C}^2(\mathbb{R}^d, \mathbb{R}_+)$ such that $\mathcal{E}(x) := \frac{1}{2}\|x - x^\star\|^2$. First, we find a bound on the infinitesimal diffusion generator of the stochastic process $\{\mathcal{E}(X(s))\}_{j\mathfrak{T} \leq s \leq (j+1)\mathfrak{T}}$:

$$\mathscr{A}\mathcal{E}(X(s)) = -\langle\nabla f(X(s)), X(s) - x^\star\rangle ds + \frac{h}{2}\text{Tr}(\Sigma_{VR}(X(s), X(s - \xi(s))))ds$$
$$\leq -\frac{\mu}{2}\|X(s) - x^\star\|^2 ds + hL^2\left(\|X(s) - x^\star\|^2 + \|X(s - \xi(s)) - x^\star\|^2\right)ds$$

where in the first inequality we used Lemma 4 and the RSI. Using Dynkin's formula (Eq. (6)), since $X(s - \xi(s)) = X(j\mathfrak{T})$ for $s \in [j\mathfrak{T}, j\mathfrak{T} + \mathfrak{T}]$ by our choice of $\xi(\cdot)$,

$$\frac{1}{2}\mathbb{E}\left[\|X(j\mathfrak{T} + \mathfrak{T}) - x^\star\|^2\right] - \frac{1}{2}\mathbb{E}\left[\|X(j\mathfrak{T}) - x^\star\|^2\right]$$
$$\leq -\frac{\mathfrak{T}}{2}(\mu - 2hL^2)\int_{j\mathfrak{T}}^{j\mathfrak{T}+\mathfrak{T}}\mathbb{E}[\|X(s) - x^\star\|^2]\frac{ds}{\mathfrak{T}} + hL^2\mathfrak{T}\mathbb{E}[\|X(j\mathfrak{T}) - x^\star\|^2],$$

which gives

$$\int_{j\mathfrak{T}}^{j\mathfrak{T}+\mathfrak{T}}\mathbb{E}[\|X(s) - x^\star\|^2]\frac{ds}{\mathfrak{T}} \leq \frac{2hL^2\mathfrak{T} + 1}{\mathfrak{T}(\mu - 2hL^2)}\mathbb{E}[\|X(j\mathfrak{T}) - x^\star\|^2].$$

By redefining (*jumping to*) $X(j\mathfrak{T} + \mathfrak{T})$ uniformly from $\{X(s)\}_{j\mathfrak{T} \leq s \leq j\mathfrak{T} + \mathfrak{T}}$, $\mathbb{E}[\|X(j\mathfrak{T} + \mathfrak{T}) - x^\star\|^2] = \int_{j\mathfrak{T}}^{j\mathfrak{T}+\mathfrak{T}}\mathbb{E}[\|X(s) - x^\star\|^2]\frac{ds}{\mathfrak{T}}$ and therefore, for all $j \in \mathbb{N}$

$$\mathbb{E}[\|X(j\mathfrak{T} + \mathfrak{T}) - x^\star\|^2] \leq \frac{2hL^2\mathfrak{T} + 1}{\mathfrak{T}(\mu - 2hL^2)}\mathbb{E}[\|X(j\mathfrak{T}) - x^\star\|^2].$$

∎

# E  Analysis in discrete-time

For ease of consultation of this appendix, we briefly describe here again our setting: $\{f_i\}_{i=1}^N$ is a collection of $L$-smooth[18] functions s.t. $f_i : \mathbb{R}^d \to \mathbb{R}$ for any $i \in \{1, \ldots, N\}$ and $f(\cdot) := \frac{1}{N}\sum_{i=1}^N f_i(\cdot)$. Trivially, $f(\cdot)$ is also $L$-smooth; our task is to find a minimizer $x^\star = \arg\min_{x \in \mathbb{R}^d} f(x)$.

**(H-)**     Each $f_i(\cdot)$ is $L$-smooth.

Mini-Batch SGD builds a sequence of estimates of the solution $x^\star$ in a recursive way, using the stochastic gradient estimate $\mathcal{G}_{\text{MB}}$:

$$x_{k+1} = x_k - \eta_k \mathcal{G}\left(\{x_i\}_{0 \leq i \leq k}, k\right), \tag{SGD}$$

where $(\eta_k)_{k \geq 0}$ is a non-increasing deterministic sequence of positive numbers called the *learning rate sequence*. We define, as in Sec. 2,

- $h := \eta_0$.
- *adjustment factor* sequence $(\psi_k)_{k \geq 0}$ s.t. for all $k \geq 0$, $\psi_k = \eta_k/h$.
- $\{\mathcal{F}_k\}_{k \geq 0}$ the natural filtration induced by the stochastic process $\{x_k\}_{k \geq 0}$.
- $\mathbb{E}$ the expectation operator over all the information $\mathcal{F}_\infty$.
- $\mathbb{E}_{\mathcal{F}_k}$ the conditional expectation given the information at step $k$.

We also report from the main paper some assumptions we might use

**(Hwqc)**     $f(\cdot)$ is $\mathcal{C}^1$ and exists $\tau > 0$ and $x^\star$ s.t. $\langle \nabla f(x), x - x^\star \rangle \geq \tau(f(x) - f(x^\star))$ for all $x \in \mathbb{R}^d$.

**(Hpl)**     $f(\cdot)$ is $\mathcal{C}^1$ and there exists $\mu > 0$ s.t. $\|\nabla f(x)\|^2 \geq 2\mu(f(x) - f(x^\star))$ for all $x \in \mathbb{R}^d$.

**(Hrsi)**     $f(\cdot)$ is $\mathcal{C}^1$ and there exists $\mu > 0$ s.t. $\langle \nabla f(x), x - x^\star \rangle \geq \frac{\mu}{2}\|x - x^\star\|^2$ for all $x \in \mathbb{R}^d$.

## E.1  Analysis of MB-SGD

### E.1.1  Non-asymptotic rates

In Sec. 2, we defined $\Sigma_{\text{MB}}(x)$ to be the one-sample conditional covariance matrix. So that $\mathbb{C}\text{ov}_{\mathcal{F}_{k-1}}[\mathcal{G}_{\text{MB}}(x_k, k)] = \frac{\Sigma_{\text{MB}}(x_k)}{b_k}$, where $b_k$ is the mini-batch size. As commonly done in the literature [22] and to match the continuous time analysis, we make the following assumption.

**(Hσ)** $\sigma_*^2 := \sup_{x \in \mathbb{R}^d} \|\sigma_{\text{MB}}(x)\sigma_{\text{MB}}(x)^T\|_S < \infty$, where $\|\cdot\|_S$ denotes the spectral norm.

Last, we define — to match existing proofs of related results [11, 22, 43], $\epsilon_k := \mathcal{G}_{\text{MB}}(x_k, k) - \nabla f(x_k)$. It follows that $\mathbb{E}[\|\epsilon_k\|^2] = \frac{d\sigma_*^2}{b_k}$.

Moreover for $k \geq 0$ we define $\varphi_{k+1} = \sum_{i=0}^k \psi_i$. We are now ready to show the non-asymptotic results. But first, we need two (well-known) classic lemmas.

---

**Lemma 5.** *Assume (H-), then*

$$\mathbb{E}[f(x_{k+1}) - f(x_k)] \leq \left(\frac{L\eta_k^2}{2} - \eta_k\right)\mathbb{E}\left[\|\nabla f(x_k)\|^2\right] + \frac{L\,d\,\sigma_*^2\,\eta_k^2}{2\,b_k}.$$

*Proof.* Thanks to the $L$-smoothness assumption, we have the classic result (see e.g. [45])

$$f(x_{k+1}) - f(x_k) \leq \langle \nabla f(x_k), x_{k+1} - x_k \rangle + \frac{L}{2} \|x_{k+1} - x_k\|^2 \qquad a.s. \qquad (12)$$

After plugging the definition of mini-batch SGD, taking the expectation and using Fubini's theorem,

$$\mathbb{E}[f(x_{k+1}) - f(x_k)]$$
$$\leq -\eta_k \mathbb{E}\left[\mathbb{E}_{\mathcal{F}_{k-1}}[\langle \nabla f(x_k), \mathcal{G}_{\mathrm{MB}}(x_k, k)\rangle]\right] + \frac{L\eta_k^2}{2}\mathbb{E}[\|\mathcal{G}_{\mathrm{MB}}(x_k, k)\|^2]$$
$$\leq -\eta_k \mathbb{E}[\langle \nabla f(x_k), \mathbb{E}_{\mathcal{F}_{k-1}}[\mathcal{G}_{\mathrm{MB}}(x_k, k)]\rangle] + \frac{L\eta_k^2}{2}\mathbb{E}[\|\nabla f(x_k) + \epsilon_k\|^2]$$
$$\leq -\eta_k \mathbb{E}\left[\|\nabla f(x_k)\|^2\right] + \frac{L\eta_k^2}{2}\mathbb{E}\left[\|\nabla f(x_k)\|^2 + \|\epsilon_k\|^2 + 2\langle \epsilon_k, \nabla f(x_k)\rangle\right]$$
$$\leq \left(\frac{L\eta_k^2}{2} - \eta_k\right)\mathbb{E}\left[\|\nabla f(x_k)\|^2\right] + \frac{L\eta_k^2}{2}\mathbb{E}_{\mathcal{F}_{k-1}}\left[\|\epsilon_k\|^2\right] + L\eta_k^2\mathbb{E}\left[\langle \mathbb{E}_{\mathcal{F}_{k-1}}[\epsilon_k], \nabla f(x_k)\rangle\right]$$
$$\leq \left(\frac{L\eta_k^2}{2} - \eta_k\right)\mathbb{E}\left[\|\nabla f(x_k)\|^2\right] + \frac{Ld\sigma_*^2\eta_k^2}{2b_k}.$$

∎

---

**Lemma 6.** *Assume (H-), then*
$$\mathbb{E}\left[\|\nabla f(x_k)\|^2\right] \leq 2L\mathbb{E}[f(x_k) - f(x^\star)].$$

---

*Proof.* We have that

$$\mathbb{E}[f(x^\star) - f(x_k)] \leq \mathbb{E}\left[f\left(x_k - \frac{1}{L}\nabla f(x_k)\right) - f(x_k)\right] \leq -\frac{1}{2L}\mathbb{E}\left[\|\nabla f(x_k)\|^2\right],$$

where the first inequality holds since $x^\star$ is the minimum and the last inequality uses Lemma 5 in the special case $\sigma_*^2 = 0$. ∎

The following theorem (statement and proof technique) has to be compared to Thm. 1 for MB-PGF.

---

**Theorem E.1.** *Assume (H-), (H$\sigma$). For $k \geq 0$ let $\tilde{k} \in [0, k]$ be a random index picked with probability $\psi_j/\varphi_{j+1}$ for $j \in \{0, \ldots, k\}$ (and 0 otherwise). If $h \leq \frac{1}{L}$, then we have:*

$$\mathbb{E}\left[\|\nabla f(x_{\tilde{k}})\|^2\right] \leq \frac{2\left(f(x_0) - f(x^\star)\right)}{(h\varphi_{k+1})} + \frac{h\,d\,L\,\sigma_*^2}{(h\varphi_{k+1})}\sum_{i=0}^{k}\frac{\psi_i^2}{b_i}h.$$

---

*Proof.* Consider the continuous-time inspired (see Thm. 1) Lyapunov function $\mathcal{E}(k) := f(x_k) - f(x^\star)$. We have, directly from Lemma 5 and using the fact that $\eta_k \leq \frac{1}{L}$ (hence $\frac{L\eta_k^2}{2} - \eta_k \leq -\frac{\eta_k}{2}$ ),

$$\mathbb{E}[\mathcal{E}(k+1) - \mathcal{E}(k)] = \mathbb{E}[f(x_{k+1}) - f(x_k)]$$
$$\leq \left(\frac{L\eta_k^2}{2} - \eta_k\right)\mathbb{E}\left[\|\nabla f(x_k)\|^2\right] + \frac{Ld\sigma_*^2\eta_k^2}{2b_k}$$
$$\leq -\frac{\eta_k}{2}\mathbb{E}\left[\|\nabla f(x_k)\|^2\right] + \frac{Ld\sigma_*^2\eta_k^2}{2b_k}$$

Finally, by linearity of integration,

$$\mathbb{E}[\mathcal{E}(k+1) - \mathcal{E}(0)] = \mathbb{E}\left[\sum_{i=0}^{k} \mathcal{E}(i+1) - \mathcal{E}(i)\right]$$

$$= \sum_{i=0}^{k} \mathbb{E}[\mathcal{E}(i+1) - \mathcal{E}(i)]$$

$$= -\frac{1}{2}\sum_{i=0}^{k} \eta_i \mathbb{E}\left[\|\nabla f(x_i)\|^2\right] + \frac{L\,d\,\sigma_*^2}{2}\sum_{i=0}^{k} \frac{\eta_i^2}{b_i}$$

$$= -\frac{h}{2}\mathbb{E}\left[\sum_{i=0}^{k} \psi_i\|\nabla f(x_i)\|^2\right] + \frac{L\,d\,h\,\sigma_*^2}{2}\sum_{i=0}^{k} \frac{\psi_i^2}{b_i} \tag{13}$$

Next, notice that, since $\sum_{i=0}^{k} \frac{\psi_i}{\varphi_{k+1}} = 1$, the function $i \mapsto \frac{\psi_i}{\varphi_{k+1}}$ defines a probability distribution. Let $\tilde{k} \in \{0, \ldots, k\}$ have this distribution; then conditioning on all the past iterations $\{x_0, \ldots, x_k\}$ and using the law of the unconscious statistician

$$\mathbb{E}_{\mathcal{F}_{k-1}}[\|\nabla f(x_{\tilde{k}})\|^2] = \frac{1}{\varphi_{k+1}}\sum_{i=0}^{k} \psi(i)\|\nabla f(x_i)\|^2 ds,$$

which, once plugged in Eq. (13), gives

$$h\varphi_{k+1}\mathbb{E}[\|\nabla f(x_{\tilde{k}})\|^2] \le 2\mathcal{E}(0) + \frac{L\,d\,h^2\,\sigma_*^2}{2}\sum_{i=0}^{k} \frac{\psi_i^2}{b_i}.$$

The proof ends by using the definition of $\mathcal{E}$. ∎

The following proposition has to be compared to Thm. 2 for MB-PGF.

---

**Theorem E.2.** *Assume (H-), (Hσ), (Hwσc) and let $\tilde{k}$ be defined as in Thm. E.1. If $0 < h \le \frac{\tau}{2L}$, then we have:*

$$\mathbb{E}\left[f(x_{\tilde{k}}) - f(x^\star)\right] \le \frac{\|x_0 - x^\star\|^2}{\tau\,(h\varphi_{k+1})} + \frac{d\,h\,\sigma_*^2}{\tau\,(h\varphi_{k+1})}\sum_{i=0}^{k} \frac{\psi_i^2}{b_i}h.$$

*Moreover, if $0 \le h \le \left(\frac{2}{L} - \frac{1}{\tau L}\right)$, then for all $k \ge 0$ we have:*

$$\mathbb{E}\left[f(x_{k+1}) - f(x^\star)\right] \le \frac{\|x_0 - x^\star\|^2}{2\,\tau\,(h\varphi_{k+1})} + \frac{h\,d\,\sigma_*^2}{2\,\tau\,(h\varphi_{k+1})}\sum_{i=0}^{k}(1 + \tau\varphi_{i+1}L)\frac{\psi_i^2}{b_i}h.$$

---

*Proof.* We prove the two rates separately.

Proof of the first formula : consider the continuous-time inspired (see Thm. 2) Lyapunov function $\mathcal{E}(k) := \frac{1}{2}\|x_k - x^\star\|^2$. We have

$$\mathbb{E}[\mathcal{E}(k+1) - \mathcal{E}(k)] =$$

$$= \frac{1}{2}\mathbb{E}\left[\|x_k - x^\star - \eta_k\mathcal{G}_{\text{MB}}(x_k, k)\|^2\right] - \frac{1}{2}\mathbb{E}\left[\|x_k - x^\star\|^2\right]$$

$$= -\eta_k\mathbb{E}\left[\mathbb{E}_{\mathcal{F}_{k-1}}\left[\langle\mathcal{G}_{\text{MB}}(x_k, k), x_k - x^\star\rangle\right]\right] + \frac{\eta_k^2}{2}\mathbb{E}\left[\|\mathcal{G}_{\text{MB}}(x_k, k)\|^2\right]$$

$$= -\eta_k\mathbb{E}\left[\langle\mathbb{E}_{\mathcal{F}_{k-1}}[\mathcal{G}_{\text{MB}}(x_k, k)], x_k - x^\star\rangle\right] + \frac{\eta_k^2}{2}\mathbb{E}\left[\|\nabla f(x_k)\|^2\right] + \frac{\eta_k^2}{2}\mathbb{E}_{\mathcal{F}_{k-1}}\left[\|\epsilon_k\|^2\right]$$

$$= -\eta_k\mathbb{E}\left[\langle\nabla f(x_k), x_k - x^\star\rangle\right] + \frac{\eta_k^2}{2}\mathbb{E}\left[\|\nabla f(x_k)\|^2\right] + \frac{d\eta_k^2\sigma_*^2}{2b_k},$$

where in the second equality we used Fubini's theorem. We proceed using weak-quasi-convexity and Lemma 6:

$$\mathbb{E}[\mathcal{E}(k+1) - \mathcal{E}(k)] \leq -\eta_k \tau \mathbb{E}\left[f(x_k) - f(x^\star)\right] + \frac{\eta_k^2}{2} \mathbb{E}\left[\|\nabla f(x_k)\|^2\right] + \frac{d\eta_k^2 \sigma_*^2}{2b_k}$$

$$\leq -\eta_k \tau \mathbb{E}\left[f(x_k) - f(x^\star)\right] + \eta_k^2 L \mathbb{E}[f(x_k) - f(x^\star)] + \frac{d\eta_k^2 \sigma_*^2}{2b_k}$$

$$\leq (L\eta_k^2 - \tau\eta_k)\mathbb{E}\left[f(x_k) - f(x^\star)\right] + \frac{d\eta_k^2 \sigma_*^2}{2b_k}.$$

Next, using the fact that $-\tau\eta_k + L\eta_k^2 \leq -\tau\frac{\eta_k}{2}$ for $\eta_k \leq \frac{\tau}{2L}$ we get

$$\mathbb{E}[\mathcal{E}(k+1) - \mathcal{E}(k)] \leq -\frac{\eta_k \tau}{2} \mathbb{E}[f(x_k) - f(x^\star)] + \frac{\eta_k^2 \sigma_*^2 d}{2b_k} \tag{14}$$

Finally, by linearity of integration,

$$\mathbb{E}[\mathcal{E}(k+1) - \mathcal{E}(0)] = \mathbb{E}\left[\sum_{i=0}^{k} \mathcal{E}(i+1) - \mathcal{E}(i)\right]$$

$$= \sum_{i=0}^{k} \mathbb{E}[\mathcal{E}(i+1) - \mathcal{E}(i)]$$

$$\leq -\frac{\tau}{2} \sum_{i=0}^{k} \eta_i \mathbb{E}\left[(f(x_i) - f(x^\star))\right] + \frac{d\sigma_*^2}{2} \sum_{i=0}^{k} \frac{\eta_i^2}{b_i}$$

$$= -\frac{\tau}{2} \mathbb{E}\left[\sum_{i=0}^{k} \eta_i (f(x_i) - f(x^\star))\right] + \frac{d\sigma_*^2}{2} \sum_{i=0}^{k} \frac{\eta_i^2}{b_i}.$$

Proceeding again as in Thm. E.1, we get the desired result.

Proof of the second formula : consider the continuous-time inspired (see Thm. 2) Lyapunov function

$$\mathcal{E}(k) := \tau h \varphi_k(f(x_k) - f(x^\star)) + \frac{1}{2}\|x_k - x^\star\|^2.$$

Then, with probability one,

$$\mathcal{E}(k+1) - \mathcal{E}(k)$$

$$= \tau h \varphi_{k+1}(f(x_{k+1}) - f(x^\star)) + \frac{1}{2}\|x_{k+1} - x^\star\|^2 - \tau h \varphi_k(f(x_k) - f(x^\star)) - \frac{1}{2}\|x_k - x^\star\|^2$$

$$= \tau h \varphi_{k+1}(f(x_{k+1}) - f(x_k)) + \tau\eta_k(f(x_k) - f(x^\star))$$

$$\quad + \frac{1}{2}\|x_k - x^\star - \eta_k \mathcal{G}_{\text{MB}}(x_k, k)\|^2 - \frac{1}{2}\|x_k - x^\star\|^2$$

$$= \tau \varphi_{k+1}(f(x_{k+1}) - f(x_k)) + \tau\eta_k(f(x_k) - f(x^\star))$$

$$\quad + \frac{\eta_k^2}{2}\|\mathcal{G}_{\text{MB}}(x_k, k)\|^2 - \eta_k\langle \mathcal{G}_{\text{MB}}(x_k, k), x_k - x^\star\rangle.$$

$$\leq \tau\eta_0 \varphi_{k+1}(f(x_{k+1}) - f(x_k)) + \tau h\eta_k(f(x_k) - f(x^\star))$$

$$\quad + \frac{\eta_k^2}{2}\|\nabla f(x_k) + \epsilon_k\|^2 - \eta_k\langle \nabla f(x_k), x_k - x^\star\rangle - \eta_k\langle \epsilon_k, x_k - x^\star\rangle.$$

$$\leq \tau h \varphi_{k+1}(f(x_{k+1}) - f(x_k)) + \frac{\eta_k^2}{2}\|\nabla f(x_k) + \epsilon_k\|^2 - \eta_k\langle \epsilon_k, x_k - x^\star\rangle,$$

where in the second equality we added and subtracted $\tau \eta_k (f(x_k) - f(x^\star))$ (recall that for $k \geq 0$, $h\varphi_{k+1} = \sum_{i=0}^{k} \eta_i$) and in the second inequality the weak-quasi-convexity assumption. Next, thanks to Lemma 5,

$$\mathbb{E}[\mathcal{E}(k+1) - \mathcal{E}(k)] =$$

$$\leq \tau h\varphi_{k+1}\mathbb{E}[f(x_{k+1}) - f(x_k)] + \frac{\eta_k^2}{2}\mathbb{E}[\|\nabla f(x_k)\|^2] + \frac{\eta_k^2}{2}\mathbb{E}[\|\epsilon_k\|^2]$$

$$= \tau h\varphi_{k+1}\mathbb{E}[f(x_{k+1}) - f(x_k)] + \frac{\eta_k^2}{2}\mathbb{E}[\|\nabla f(x_k)\|^2] + \frac{\eta_k^2}{2}\mathbb{E}_{\mathcal{F}_{k-1}}[\|\epsilon_k\|^2]$$

$$\leq \tau h\varphi_{k+1}\mathbb{E}[f(x_{k+1}) - f(x_k)] + \frac{\eta_k^2}{2}\mathbb{E}\|\nabla f(x_k)\|^2 + \frac{\eta_k^2 d\sigma_*^2}{2b_k}$$

$$\leq \tau h\varphi_{k+1}\left(\left(\frac{L\eta_k^2}{2} - \eta_k\right)\mathbb{E}\left[\|\nabla f(x_k)\|^2\right] + \frac{Ld\sigma_*^2\eta_k^2}{2b_k}\right) + \frac{\eta_k^2}{2}\mathbb{E}[\|\nabla f(x_k)\|^2] + \frac{\eta_k^2 d\sigma_*^2}{2}$$

$$\leq \left(\frac{\eta_k^2}{2} + \tau\eta_0\varphi_{k+1}\left(\frac{L\eta_k^2}{2b_k} - \eta_k\right)\right)\mathbb{E}\left[\|\nabla f(x_k)\|^2\right] + \frac{\eta_k^2 d\sigma_*^2(1 + L\tau\varphi_{k+1})}{2b_k}.$$

If $h \leq 2/L$, then $\frac{L\eta_k^2}{2} - \eta_k \leq 0$. Moreover, under this condition, since for all $k \geq 0$ we have $\varphi_{k+1} \geq \eta_k$, it is clear that $\varphi_{k+1}\left(\frac{L\eta_k^2}{2} - \eta_k\right) \leq \eta_k\left(\frac{L\eta_k^2}{2} - \eta_k\right)$. Hence

$$\mathbb{E}[\mathcal{E}(k+1) - \mathcal{E}(k)] \leq \left(\frac{\eta_k^2}{2} + \tau\eta_k\left(\frac{L\eta_k^2}{2} - \eta_k\right)\right)\mathbb{E}\left[\|\nabla f(x_k)\|^2\right] + \frac{\eta_k^2 d\sigma_*^2(1 + L\tau\varphi_{k+1})}{2b_k}.$$

It is easy to see that $\frac{\eta_k^2}{2} + \tau\eta_k\left(\frac{L\eta_k^2}{2} - \eta_k\right) \leq 0$ if and only if $h \leq \frac{2\tau - 1}{\tau L}$. Under this condition, since $\mathbb{E}\left[\|\nabla f(x_k)\|^2\right] \geq 0$,

$$\mathbb{E}[\mathcal{E}(k+1) - \mathcal{E}(k)] \leq \frac{\eta_k^2 d\sigma_*^2(1 + L\tau\varphi_{k+1})}{2b_k}.$$

Finally, by linearity of integration,

$$\mathbb{E}[\mathcal{E}(k+1) - \mathcal{E}(0)] = \mathbb{E}\left[\sum_{i=0}^{k} \mathcal{E}(i+1) - \mathcal{E}(i)\right]$$

$$= \sum_{i=0}^{k} \mathbb{E}[\mathcal{E}(i+1) - \mathcal{E}(i)] = \frac{d\sigma^2 h^2}{2}\sum_{i=0}^{k}\frac{\psi_i^2(1 + L\tau\varphi_{i+1})}{b_i}.$$

The result then follows from the definition of $\mathcal{E}$. ∎

The following proposition has to be compared to Thm. 3 for MB-PGF.

**Theorem E.3.** *Assume (H), (Hσ), (HPL). If $h \leq 1/L$, then for all $k \geq 0$ we have:*

$$\mathbb{E}\left[(f(x_{k+1}) - f(x^\star))\right] \leq$$

$$\left(\prod_{i=0}^{k}(1 - \mu\,h\psi_i)\right)(f(x_0) - f(x^\star)) + \frac{h\,d\,L\,\sigma_*^2}{2}\sum_{i=0}^{k}\frac{\prod_{\ell=0}^{k}(1 - \mu\,h\psi_\ell)}{\prod_{j=0}^{i}(1 - \mu\,h\psi_l)}\frac{\psi_i^2}{b_i}h.$$

*Proof.* Starting from Lemma 5 we apply the PŁ property. If $\frac{L\eta_k^2}{2} - \eta_k \leq 0$, that is $\eta_k \leq 2/L$ for all $k$, then

$$\mathbb{E}[f(x_{k+1}) - f(x_k)] \leq \left(\frac{L\eta_k^2}{2} - \eta_k\right) \mathbb{E}\left[\|\nabla f(x_k)\|^2\right] + \frac{Ld\sigma_*^2\eta_k^2}{2}$$

$$\leq 2\mu\left(\frac{L\eta_k^2}{2} - \eta_k\right) \mathbb{E}[f(x_k) - f(x^\star)] + \frac{Ld\sigma_*^2\eta_k^2}{2}$$

Furthermore, if $\eta_k \leq 1/L$ for all $k$ then $\frac{L\eta_k^2}{2} - \eta_k \leq -\frac{\eta_k}{2b_k}$:

$$\mathbb{E}[f(x_{k+1}) - f(x_k)] \leq -\mu\eta_k \mathbb{E}[f(x_k) - f(x^\star)] + \frac{Ld\sigma_*^2\eta_k^2}{2b_k}. \tag{15}$$

Consider now the Lyapunov function inspired by the continuous time prospective (see Thm. 3):

$$\mathcal{E}(k) := \begin{cases} \prod_{i=0}^{k-1}(1 - \eta_i\mu)^{-1}(f(x_k) - f(x^\star)) & k > 0 \\ (f(x_k) - f(x^\star)) & k = 0 \end{cases}.$$

We have, for $k \geq 0$,

$$\mathbb{E}[\mathcal{E}(k+1) - \mathcal{E}(0)]$$

$$= \mathbb{E}\left[\sum_{i=0}^{k} \mathcal{E}(i+1) - \mathcal{E}(i)\right]$$

$$= \sum_{i=0}^{k} \mathbb{E}[\mathcal{E}(i+1) - \mathcal{E}(i)]$$

$$= \sum_{i=0}^{k}\left(\prod_{j=0}^{i}(1 - \eta_j\mu)^{-1}\right) \mathbb{E}\left[f(x_{i+1}) - f(x^\star) - (1 - \eta_i\mu)(f(x_i) - f(x^\star))\right]$$

$$= \sum_{i=0}^{k}\left(\prod_{j=0}^{i}(1 - \eta_j\mu)^{-1}\right) \mathbb{E}\left[f(x_{i+1}) - f(x_i) + \eta_i\mu(f(x_i) - f(x^\star))\right].$$

Using Lemma 5,

$$\mathbb{E}[\mathcal{E}(k+1) - \mathcal{E}(0)]$$

$$\leq \sum_{i=0}^{k}\left(\prod_{j=0}^{i}(1 - \eta_j\mu)^{-1}\right)\left(-\mu\eta_i\mathbb{E}[f(x_i) - f(x^\star)] + \frac{Ld\sigma_*^2\eta_i^2}{2b_i} + \eta_i\mu\mathbb{E}[f(x_k) - f(x^\star)]\right)$$

$$\leq \sum_{i=0}^{k}\left(\prod_{j=0}^{i}(1 - \eta_j\mu)^{-1}\right)\frac{Ld\sigma_*^2\eta_i^2}{2b_i},$$

where in the first inequality we used Eq. (15) . By plugging in the definition of $\mathcal{E}$,

$$\prod_{i=0}^{k}(1 - \eta_i\mu)^{-1}\mathbb{E}_{\mathcal{F}_k}\left[(f(x_{k+1}) - f(x^\star))\right] \leq f(x_0) - f(x^\star) + \sum_{i=0}^{k}\left(\prod_{j=0}^{i}(1 - \eta_j\mu)^{-1}\right)\frac{Ld\sigma_*^2\eta_i^2}{2b_i}$$

which gives the desired result. ∎

| Condition | Limit | Bound |
|---|---|---|
| **(H-)**, **(Hσ)** | $\lim_{k\to\infty} \mathbb{E}\left[\|\nabla f(x_{\tilde{k}})\|^2\right]$ | $\frac{L\,d\,\sigma_*^2\,h}{b}$ |
| **(H-)**, **(Hσ)**, **(HwQC)** | $\lim_{k\to\infty} \mathbb{E}\left[f(x_{\tilde{k}}) - f(x^\star)\right]$ | $\frac{L\,d\,\sigma_*^2\,h}{\tau\,b}$ |
| **(H-)**, **(Hσ)**, **(HPŁ)** | $\lim_{k\to\infty} \mathbb{E}\left[f(x_k) - f(x^\star)\right]$ | $\frac{L\,d\,\sigma_*^2 h}{2\,\mu\,b}$ |

Table 5: Ball of convergence of MB-PGF under constant $\psi_k = 1$ and $b_k = b$. For $k \geq 0$, $\varphi_{k+1} = \sum_{i=0}^{k} \psi_i$ and $\tilde{k} \in [0, k]$ is a random index picked with distribution $\psi_j / \varphi_{j+1}$ for $j \in \{0, \ldots, k\}$ and 0 otherwise.

### E.1.2 Asymptotic rates under decreasing adjustment factor

Can be derived easily using the same arguments as in App. D.3, with the same final results.

### E.1.3 Limit sub-optimality under constant adjustment factor

In this paragraph we pick $\psi_k = 1$ and $b_k = b$ for all $k$ and study the ball of convergence of SGD. The results can be found in Tb. 5. The only non-trivial limit is the one for PŁ functions.

By direct calculation.

$$\mathbb{E}\left[(f(x_{k+1}) - f(x^\star))\right] \leq (1 - h\mu)^{k+1}(f(x_0) - f(x^\star)) + \frac{Ld\sigma_*^2 h^2}{2b}\sum_{i=0}^{k}(1 - h\mu)^i$$

$$\leq (1 - h\mu)^{k+1}(f(x_0) - f(x^\star)) + \frac{Ld\sigma_*^2 h^2}{2b}\sum_{i=0}^{\infty}(1 - h\mu)^i$$

$$= (1 - h\mu)^{k+1}(f(x_0) - f(x^\star)) + \frac{Ld\sigma_*^2 h^2}{2hb\mu}.$$

Where we used the fact that for any $\rho < 1$, $\sum_{i=0}^{\infty} \rho_i = \frac{1}{1-\rho}$. The result then follows taking the limit.

### E.1.4 Convergence rates for VR-SGD (SVRG)

**Theorem E.4.** *Assume (H-), (HRSI) and choose $\xi_k = k - \sum_{j=1}^{\infty} \delta_{k-jm}$ (sawtooth wave), where $\delta$ is the Kronecker delta. Let $\{x_k\}_{k\geq 0}$ be the solution to SGD with VR with additional jumps at times $(jm)_{j\in\mathbb{N}}$: we jump picking $x_{(j+1)m}$ uniformly in $\{x_k\}_{jm\leq k<(j+1)m}$. Then,*

$$\mathbb{E}[\|x_{jm} - x^\star\|^2] = \left(\frac{1 + 2L^2h^2m}{hm(\mu - 3L^2h)}\right)^j \|x_0 - x^*\|^2.$$

*Proof.* Start by computing

$$\frac{1}{2}\mathbb{E}\left[\|x_{k+1} - x^\star\|^2\right]$$

$$= \frac{1}{2}\mathbb{E}\left[\|x_k - x^\star - h\mathcal{G}_{\text{VR}}(k)\|^2\right]$$

$$= -h\mathbb{E}\left[\langle\nabla f(x_k), x_k - x^\star\rangle\right] + L^2h^2\mathbb{E}[\|\mathcal{G}_{\text{VR}}(k)\|^2]$$

where we used the fact that $\mathcal{G}_{\text{VR}}$ is unbiased. Consider iterations $jm \leq k \leq j(m+1)$. Our choice of $\xi$ fixes the pivot to $x_{jm}$. Using smoothness, Cor. 4 and the restricted-secant-inequality, we get,

$$
\begin{aligned}
&\frac{1}{2}\mathbb{E}\left[\|x_{k+1} - x^\star\|^2\right] - \frac{1}{2}\mathbb{E}\left[\|x_k - x^\star\|^2\right] \\
&\leq -h\mathbb{E}\left[\langle\nabla f(x_k), x_k - x^\star\rangle\right] + L^2 h^2 \mathbb{E}\left[\|x_k - x^\star\|^2\right] + L^2 h^2 \mathbb{E}\left[\|x_{jm} - x^\star\|^2\right] \\
&\leq -\frac{h\mu}{2}\mathbb{E}\left[\|x_k - x^\star\|^2\right] + 2L^2 h^2 \mathbb{E}\left[\|x_k - x^\star\|^2\right] + L^2 h^2 \mathbb{E}\left[\|x_{jm} - x^\star\|^2\right] \\
&= -\frac{h}{2}\left(\mu - 2L^2 h\right)\mathbb{E}\left[\|x_k - x^\star\|^2\right] + L^2 h^2 \mathbb{E}\left[\|x_{jm} - x^\star\|^2\right].
\end{aligned}
$$

Finally, summing from $jm$ to $j(m+1)$, we have

$$
\begin{aligned}
&\frac{1}{2}\mathbb{E}\left[\|x_{j(m+1)} - x^\star\|^2\right] - \frac{1}{2}\mathbb{E}\left[\|x_{jm} - x^\star\|^2\right] \\
&\qquad\qquad \leq -\frac{hm}{2}\left(\mu - 2L^2 h\right)\frac{1}{m}\sum_{k=jm}^{jm+m-1}\mathbb{E}\left[\|x_k - x^\star\|^2\right] + L^2 h^2 m \mathbb{E}\left[\|x_{jm} - x^\star\|^2\right].
\end{aligned}
$$

Therefore, dropping the first term,

$$
\mathbb{E}\left[\|x_{(j+1)m} - x^\star\|^2\right] = \frac{1}{m}\sum_{k=jm}^{jm+m-1}\mathbb{E}\left[\|x_k - x^\star\|^2\right].
$$

Redefining (*jumping* to) $x_{j(m+1)} \sim \mathcal{U}(\{x_k\}_{jm \leq k \leq (j+1)m})$, we get

$$
\mathbb{E}[\|x_{j(m+1)} - x^\star\|^2] \leq \frac{1 + 2L^2 h^2 m}{hm(\mu - 2L^2 h)}\mathbb{E}\left[\|x_{jm} - x^\star\|^2\right].
$$

$\blacksquare$

# F Time stretching

> **Theorem F.1** (Restated Thm. 5). *Let* $\{X(t)\}_{t \geq 0}$ *satisfy PGF and define* $\tau(\cdot) = \varphi^{-1}(\cdot)$, *where* $\varphi(t) = \int_0^t \psi(s)ds$. *For all* $t \geq 0$, $X(\tau(t)) = Y(t)$ *in distribution, where* $\{Y(t)\}_{t \geq 0}$ *satisfies*
>
> $$
> dY(t) = -\nabla f(Y(t))dt + \sqrt{\frac{h\,\psi(\tau(t))}{m(\tau(t))}}\sigma(\tau(t))\,d\tilde{B}(t),
> $$
>
> *where* $\{\tilde{B}(t)\}_{t \geq 0}$ *is a Brownian Motion.*

*Proof.* By definition, $X(t)$ is such that

$$
X(t) = -\int_0^t \psi(r)\nabla f(X(r))\,dr + \int_0^t \psi(r)\sqrt{\frac{h}{m(r)}}\sigma(r)\,dB(r).
$$

Therefore

$$
X(\tau(t)) = \underbrace{-\int_0^{\tau(t)} \psi(r)\nabla f(X(r))\,dr}_{:=A} + \underbrace{\int_0^{\tau(t)} \psi(r)\sqrt{\frac{h}{m(r)}}\sigma(r)\,dB(r)}_{:=B}.
$$

Using the change of variable formula for Riemann integrals, we get

$$
A = \int_0^{\tau(t)} \psi(r)\nabla f(X(r))dr = -\int_0^t \tau'(r) \cdot \psi(\tau(r)) \cdot \nabla f(X(\tau(r)))dr =
$$

$$
= \int_0^t \frac{\cancel{\psi(\tau(r))}}{\cancel{\psi(\tau(r))}}\nabla f(X(\tau(r)))dr.
$$

Using the time change formula (Thm. B.4) for stochastic integrals, with $v(r) := \psi(r)\sqrt{\frac{h}{m(r)}}\sigma(r)$,

$$B = \int_0^{\tau(t)} \psi(r)\sqrt{\frac{h}{m(r)}}\sigma(r)\,dB(r) = \int_0^t \frac{\psi(\tau(r))}{\sqrt{\tau'(r)}}\sqrt{\frac{h}{m(\tau(r))}}\sigma(\tau(r))\,d\tilde{B}(r) =$$

$$= \int_0^t \sqrt{\frac{h\,\psi(\tau(r))}{m(\tau(r))}}\sigma(\tau(r))\,d\tilde{B}(r).$$

All in all, we have found that

$$X(\tau(t)) = -\int_0^t \nabla f(X(\tau(r)))dr + \int_0^t \sqrt{\frac{h\,\psi(\tau(r))}{m(\tau(r))}}\sigma(\tau(r))\,d\tilde{B}(r).$$

By Def. 2, this is equivalent to saying that $Y := X \circ \tau$ satisfies the differential in the theorem statement. ∎

## Footnotes

[16]Because $\langle \partial_x \mathcal{E}(X(t), t), \sigma(t) \rangle \in \mathcal{M}^2([0, T], \mathbb{R})$, see e.g. Thm. 1.5.8 [41]

[17] $\varphi(t)$ is the integral of $\psi(t)$, which starts positive, so it is positive for $t > 0$.

[18]As already mentioned in the main paper, we say a function $f \in \mathcal{C}^1(\mathbb{R}^d, \mathbb{R}^m)$ is $L$-smooth if, for all $x, y \in \mathbb{R}^d$, we have $\|\nabla f(x) - \nabla f(y)\| \leq L\|x - y\|$