[Reviews · NeurIPS 2019]

Reviewer 1



I enjoyed reading the paper immensely-thank you very much for the effort made to provide intuition and to link your work to both recent results as well as to classic techniques (like the ODE method). I have very little to ask here. As a question of more academic nature: the Gaussianity assumption makes perfect sense, and the explanation provided in 117-126 is more than sufficient. It is also great that intuition via the smoothness of the objective is given as the reason as to why an "algebraic" relationship exists between the rate proofs for the discrete and continuous cases. That said, I believe that it could be possible to make both of these concepts precise, i.e., establish a form of "mean field limit": scaling n, b, and h jointly one may be able to show that the resulting rescaled discrete trajectory converges to the trajectory of the SDE in a precise manner. Such "mean field" results exist for the ODE method (see, e.g., https://doi.org/10.1016/j.peva.2008.03.005 and https://doi.org/10.1017/S0143385798097557). I do believe that this may be a whole lot lot of work to prove something that one is already inclined to believe (in the limit, the LLN kicks in!), and this sense this effort may not be worth it-and I am not advocating that this is needed for the present submission. It may be however a future direction worth pursuing even in a simpler setting than MB-SGD or SVRG, that could potentially make the link between the discrete approach and the continuous approach both formal and airtight.

Reviewer 2



This paper builds continuous-time models for SGD and stochastic variance reduction methods, by providing explicit expressions of the evolution equations. Based on these expressions, this paper analyzes the convergence rates of the continuous-time models under certain conditions, and shows strong connections in convergence rate between the continuous-time model and its discrete-time counterpart (SGD or SVRG). Strengths: > The paper successfully builds continuous-time models that take into consideration both a) decaying step-size and b) growing batch-size of the corresponding discrete-time algorithms. And it shows the existence and uniqueness of the continuous-time models. > The paper provides convergence rates of the continuous-time models, under certain conditions. Convergence rates for the corresponding discrete-time counterparts are also developed. Weaknesses: >The concept of building continuous-time models (or ODEs) for stochastic optimization algorithms is not novel. Simpler continuous-time models for SGD can be found in the reference [47] [48] of the paper, where the stochasticity of gradient is also represented by Brownian motion. The continuous-time models built in this paper can be seen as extensions of the model in [47]. >Building the continuous-time models in this paper is not technically hard. Compared to the models in [47][48], the models in this paper additionally incorporate time-dependent factors that originate from decaying step-size and growing batch-size, which can be done by simply adding two time-dependent functions.

Reviewer 3



I have read the rebuttal and I believe the authors have satisfactorily addressed my comments on prior work, so I have increased my rating. =========== Reasons for contribution ratings: 1. The SDE approximation method is well-established. Moreover, Minibatch SGD's continuous approximation has been considered by several prior works, e.g. [A] below. 2. Matching bounds are obtained, which is interesting 3. The time-change observation is quite immediate, but landscape change is non-trivial and potentially significant -- although I do have some doubts on its validity in general (more below). Summary and review comments: The paper is well-written and one of its strengths in generally good comparison with prior work. The main theoretical results are: * SDE approximation for minibatch SGD and SVRG * Well-posedness of the SDEs * Matching convergence bounds using Lyapunov functions * Interpreting time-dependent adjustments as time-change and landscape-stretching. The main weakness is the lack of focus of the discussion. I feel that too many points are scattered and there lacks a central message on the insights gained. Below are some specific questions and concerns: 1. Line 100-101: The theoretical results in [36] and also [26] do not assume that the gradient noise $Z_k$ is Gaussian. The weak approximation results do not depend on the actual distribution of gradient noise, which only need to satisfy some moment conditions. These are always satisfied when the objective is of finite-sum form, as considered in this work. See also [B] below for more general statements. This part should be rephrased accordingly to properly represent the results of prior work. 2. Line 180: The assumption $H\sigma$ is quite restrictive, as even in the quadratic case, as long as the covariance of gradients are not constant you would expect there to be some growth. I suggest relaxing this condition by some localization arguments, since at the end your results only depend on $\sigma^*$. 3. Line 127-137: 1) The reference appears wrong, [37] does not talk about the convergence rate of SGD to SDE. 2) Note that in previous work, explicit bounds between expectations over arbitrary test functions (not just $||\nabla f||^2$) on SGD and SDE are established. These are not the same as the results presented in Appendix D, which are matching rates just on $||\nabla f||^2$ (not arbitrary test functions). Moreover, the presented results are not bounding the difference between the expectation iterates, but rather show them having similar rates. This is a weaker statement. In my opinion, this point should be better clarified to avoid confusion of what actualy is derived in this paper -- in fact, without looking at the appendix I thought that the authors obtained uniform-in-time approximation results for non-convex cases, which would certainly be interesting! As far as I know, so far only [C] provides such estimates, but require strong convexity. I suggest the authors make space for the statements of results in this section in the main paper, since you have mentioned this in your abstract as one of your main results. 4. Line 277-286: This is an interesting observation. However, I have some concerns on its validity in general settings. It is well-known that 1D SDEs with multiplicative noise can be written as a noisy gradient flow of a modified potential function, but this fails to hold in high dimensions. It appears to me that by assuming $H$ is diagonal and $\sigma$ is constant, we fall into the 1D scenario, but this analogy is not likely to generalize. Perhaps the authors can comment on this. 5. Minor typos: 1) Theorem B.2, assumption 1 should not have a square on the RHS. 2) line 194: know -> known References: [A] Smith, Samuel L., and Quoc V. Le. "A bayesian perspective on generalization and stochastic gradient descent." arXiv preprint arXiv:1710.06451 (2017). [B] Li et al. "Stochastic Modified Equations and Dynamics of Stochastic Gradient Algorithms I: Mathematical Foundations." Journal of Machine Learning Research 20.40 (2019): 1-40. [C] Feng, Yuanyuan, et al. "Uniform-in-Time Weak Error Analysis for Stochastic Gradient Descent Algorithms via Diffusion Approximation." arXiv preprint arXiv:1902.00635 (2019).

[Author Response · NeurIPS 2019]

We thank all the reviewers for their valuable feedback. We address their individual concerns below.

## Reviewer 1

We are grateful for the very positive comments on our work and for the outlined valuable directions for future research. We totally agree on the need to establish a more rigorous connection between the continuous- and discrete-time perspectives in optimization — this is exactly the focus of our current research: we see this paper as our first step in this direction. We will include a discussion of the references you indicate in your review, including the mean field limit work you mentioned which is related to our interest for future research directions.

## Reviewer 2

**Models are easy extensions of existing ones**   As correctly stated by the reviewer (and actually also by us in the paper), SDE approximations of SGD (with fixed learning rate) were already presented in previous literature (see e.g. line 151). In this paper we extend these models to include the effect of decaying learning rates, increasing batch-sizes and variance reduction. We agree that the construction of such models, to the eyes of an expert reader, might not be technically very challenging — but this is precisely why *we dedicate only one page to this part* before switching our focus to the convergence analysis. Moreover, we would like to point out that

1. studying such extensions is of chief importance in machine learning, since modern SGD algorithms (see e.g. [41], [7], [29]) rely on decaying learning rates, increasing batch-sizes and variance reduction;
2. the construction of the model is *just one of the 5 contributions* listed in our paper. As stated in the introduction, our main goal is to show how such model can guide the analysis of commonly used stochastic methods.

**Possible extension to continuous-time models for momentum-based accelerated method**   Some continuous-time stochastic momentum SDEs have been studied in [32] — a truly beautiful work from which we took a lot of inspiration. However, this work is of a different nature: it does not focus on providing a correspondence between continuous models and discrete algorithms; but instead analyses a very general yet interpretable SDE in the *convex* setting. That said, we agree on the need to extend our methodology to stochastic momentum methods: we have in fact already derived similar results, but decided not to include it in our submission to focus more on our other contributions. Indeed, as noted by R1, the paper is already quite dense and the study of momentum methods *deserves to be explored in a separate work*.

## Reviewer 3

We apologize for the typos/wrong reference numbers, and we thank the reviewer for pointing them out: [36] at line 101 should be [40], [37] at line 127 should be [36].

**Assumption (H$\sigma$) restrictive (comment 2)**   We would first like to point out that such assumption is commonly used in the continuous-time literature (see e.g. (H$_3$) in [40] and Eq.(8) in [32,NeurIPS Proceedings]). Nevertheless, we thank the reviewer for the constructive comment and *will try to update our proofs* using the suggested localization argument.

**Generality of landscape stretching result (comment 4)**   We thank the reviewer for the interesting comment. We believe that the landscape stretching phenomenon is actually quite general and would also hold e.g. asymptotically under strong convexity[1]: indeed it is well known that, by Taylor's theorem, in the neighborhood of the solution to a strongly convex problem the cost *behaves as its quadratic approximation*. In dynamical systems, this linearization argument can be made very precise and goes under the name of Hartman-Grobman theorem. Since the process we study is memoryless (no momentum), at some point it will necessarily enter a neighborhood of the solution where the dynamics is described by the landscape stretching result. We will add a comment on this in the updated version of this paper. We are also thankful for the comment about multiplicative noise in high dimensions; yet, at least in the simple case we considered, the volatility is not actually a function of the state. We will nevertheless add a short note on this.

**Weak approximation of SGD (comment 1 and 3)**   *We clearly stated at line 129* that [26] and [36] do not assume Gaussianity of $Z_k$. We are very fund of these works (including [B]), yet we find that practical implications are limited since the approximation bound explodes (as the standard Euler global error does) as a function of the final time point. That is why we explicitly said (line 135) that *our approach in this paper is different*: we explore the connection between continuous and discrete exclusively by providing matching rates (and the algebraic equivalence in Sec 3.2 and App. A.2.). We are very sorry for the confusion, but we felt this point was clear given that it is also stressed many times in Sec. 3.2 as well as in the list of contributions (line 59) and in the abstract (line 6). Nonetheless, we understand that this is a crucial point and *we will do our best to update line 127-137* to make sure there is absolutely no confusion in where our contribution lies with respect to prior work. We also thank the reviewer for pointing out [C] to us (very interesting work), which we will also include in the discussion.

**In summary** we would benefit from the additional ninth page to extend and update the discussion as outlined above. In particular, to improve transparency, we would *transfer details from App. D* to the main paper as the reviewer suggests.

## Footnotes

[1]and also in the neighborhood of any hyperbolic fixed point, with implications about saddle point evasion.


[Meta-Review · NeurIPS 2019]

The paper presents an SDE approximation of mini-batch stochastic gradient descent and stochastic variance reduction gradient descent, two widely used methods, and they derive convergence rates. Well written paper. It presents a nice (i.e., not revolutionary, but still of interest to the community) result that fits within this area. Reviewers have a few suggestions for clarifications/improvements. [This meta-review was reviewed and revised by the Program Chairs]